# Conventional RVS Methods for Seismic Risk Assessment for Estimating the Current Situation of Existing Buildings: A State-of-the-Art Review

**Nurullah Bektaş *** and **Orsolya Kegyes-Brassai**

Department of Structural Engineering and Geotechnics, Széchenyi István University, 9026 Győr, Hungary; kegyesbo@sze.hu
* Correspondence: nurullahbektas@hotmail.com

**Abstract:** Developments in the field of earthquake engineering over the past few decades have contributed to the development of new methods for evaluating the risk levels in buildings. These research methods are rapid visual screening (RVS), seismic risk indexes, and vulnerability assessments, which have been developed to assess the levels of damage in a building or its structural components. RVS methods have been proposed for the rapid pre- and/or post-earthquake screening of existing large building stock in earthquake-prone areas on the basis of sidewalk surveys. The site seismicity, the soil type, the building type, and the corresponding building characteristic features are to be separately examined, and the vulnerability level of each building can be identified by employing the RVS methods. This study describes, evaluates, and compares the findings of previous investigations that utilized conventional RVS methods within a framework. It also suggests the methods to be used for specific goals and proposes prospective enhancement strategies. Furthermore, the article discusses the time-consuming RVS methods (such as FEMA 154, which requires from 15 to 30 min, while NRCC requires one hour), and provides an overview of the application areas of the methods (pre-earthquake: FEMA 154, NRCC, NZEE, etc.; postearthquake: GNDT, EMS, etc.). This review of the traditional RVS methods offers a comprehensive guide and reference for field practitioners (e.g., engineers, architects), and recommends enhancement techniques (e.g., machine learning, fuzzy logic) for researchers to be used in future improvements.

**Keywords:** building vulnerability; rapid visual screening; earthquakes; vulnerability assessment; damage grade; risk reduction

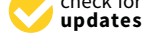



## 1. Introduction

Unfortunately, an earthquake-prone area with a long return period of main seismic activity causes an insufficient consideration to preparations against the seismic actions in low- or medium-seismic-prone areas. This has led to nonseismic designs that are created without consideration of impending earthquakes [1,2]. According to Bari [3], despite the need for new buildings because of the economic crises of European countries, the number of new buildings to be constructed has decreased; i.e., there is an aging building stock. The structural seismic vulnerability is characterized as the system sensitivity to damage that is caused by ground shaking of a certain intensity [4]. The objective of the seismic vulnerability assessments is to determine the likelihood of there being a certain amount of damage to a certain form of building because of an impending earthquake [5]. Seismic vulnerability assessment methodologies are widely used to identify and manage the risks of building and infrastructure damage, as well as the economic losses, in the event of a seismic hazard or hypothetical earthquake, because some of the settlements are in seismically prone areas. The structural assessment is based on a three-stage vulnerability evaluation, which includes rapid visual screening (RVS), a preliminary vulnerability assessment (PVA), and a detailed vulnerability assessment (DVA). In the case of each building, the overall

methodology should be followed to prepare detailed project reports, as is shown in Figure 1, which was developed by the Applied Technology Council (ATC).

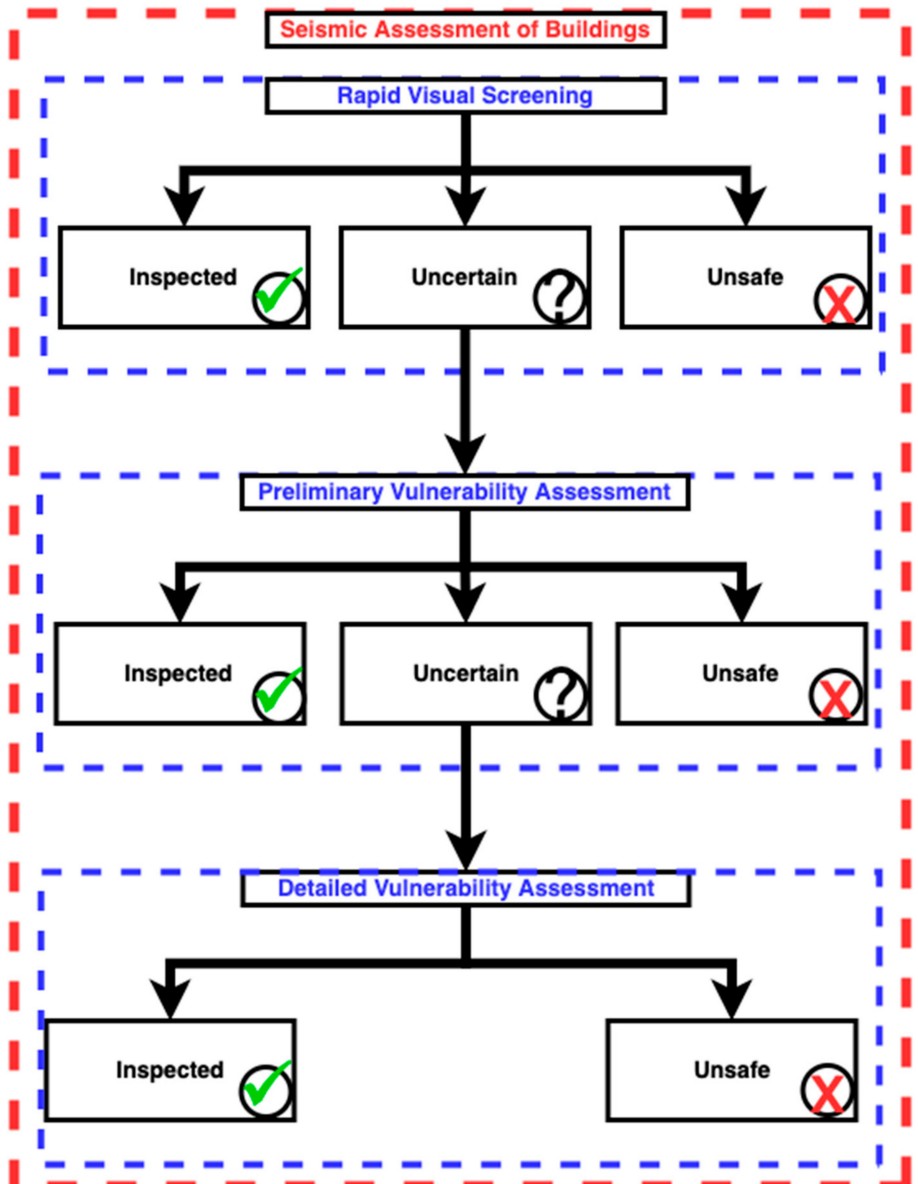

**Figure 1.** Flowchart for normal building safety evaluation and posting (Compiled by authors based on [6]).

RVS, the first stage of the seismic damage assessment methods, with 1–2-page survey forms [7], consists of the seismic sensitivity assessment process to define the structural integrity and reliability of the buildings through inspection. RVS is a visual examination method for buildings and their components that is performed by an experienced screener through a sidewalk survey. RVS methods are generally used to assess buildings without any calculations. Moreover, RVS uses a quick scan to estimate the losses that could occur in a building because of an imminent earthquake. A wide variety of traditional RVS methods (FEMA 154 [8–10]; EMS-98 [11]; the RISK-UE Project [12]; OASP [13]; NRCC [14], etc.) have been developed in recent years. Some of the structures were built before the current standards, which indicates that these buildings are potentially vulnerable to heavy seismic impacts. Therefore, these rapid assessment methodologies are important in the case of the present structures. RVS methods are widely used to determine the seismic vulnerability of an individual building or a large building stock in a city on the basis of visual inspection,

without any structural calculations. Since RVS approaches are based on visual inspection, they have the major advantage of being able to assess each building in a relatively short period of time, and they are applicable for both pre- and postearthquake seismic risk mitigation. Therefore, this quick scan saves time and resources [15,16]. An inventory of the characteristic building features (such as the structural and nonstructural systems, the design code, the plan and vertical irregularities, the seismic zone, and the site soil characteristics) is used for the visual building examination. On the basis of the visually collected data, the weight of each affecting parameter is taken into account in a numerical scoring system for the seismic vulnerability. Finally, the overall performance score is obtained.

At the RVS stage, the structures that do not meet the required expectations will be handled with more detailed assessment methods. The buildings not achieving a threshold value in the RVS are subjected to a second stage of evaluation, which is the PVA. The second stage consists of a more comprehensive analysis of the various building components, such as the site soil conditions, the properties of the materials used, the condition of the structural elements, as well as the preparation of the drawings and the load calculations. Structural drawings are prepared, and the loads are calculated in order to perform simplified analyses that are based on the various methods of the simplified structural model.

The third comprehensive seismic evaluation stage is the final stage, and it is applied as a result of a building being determined as insufficient on the basis of the first two evaluation stages [17]. This process, which is known as a "structural seismic assessment", consists of a retail assessment of each building component, and linear and nonlinear analyses using the finite element method (FEM), the applied element method (AEM), and vulnerability and fragility assessment methodologies [18]. At this point, several elastic analysis methods (e.g., linear static analysis, linear dynamic analysis, method of complex response, etc.) and/or archetype simplified structural modellings are performed in the literature [19–24] in order to examine the fragilities of the buildings. Furthermore, nonlinear static (pushover analysis, the N2 method, etc.) and nonlinear dynamic (incremental dynamic analysis (IDA), the endurance time (ET) method, etc.) analyses are performed for the fragility assessments. A static examination of how far along the inelastic zone a structure may drift before experiencing global instability is known as a "pushover analysis". Because of the fact that earthquake excitations vary significantly, researchers [19,25–28] have employed static pushover analyses to determine the structural responses [29]. Furthermore, the ET method, a detailed seismic assessment methodology, is implemented on the basis of an artificial ramp-type standard acceleration record of at least 10 s to the considered engineering structure, until the maximum structural response value is obtained [30,31]. The time that the damage limit index is exceeded is used to identify the structural performance [30,32]. Numerous studies [31,33–35] in the literature have employed the ET method for comprehensive building vulnerability assessments. The IDA approach, the most detailed structural assessment approach, was developed as a method for determining the relationship between the structural failure and the intensity measures [36,37]. Subsequently, this approach has been used as the basis for a number of standards (such as FEMA P-695 [38]) and by researchers [39–46] for the assessment of the hazard level of the structural model. In addition, a number of studies [47–50] have been conducted with the aim of strengthening the engineering structures.

Some of the existing buildings could be seismically vulnerable because they were constructed without consideration to the seismic design standards, or before the seismic design regulations or the creation of more enhanced versions, as well as because of inappropriate building construction. Building destruction and damage during an impending earthquake have detrimental impacts on economic development, natural resource protection, and economic revenue. Therefore, existing buildings should be assessed to evaluate the structural safety according to the most recently published seismic regulations. Thus, existing conventional RVS procedures should be well known so that appropriate modifications and/or developments can be performed to overcome the weak spots (e.g., the accuracy, containing the site-specific building characteristics). This study presents a comprehensive state-of-the-

art review of the currently used traditional RVS methods, and it offers a comprehensive guide and reference for a broad audience, including structural engineers, design professionals, architects, building officials, construction contractors, researchers, insurance companies, and nonprofessionals. Our study describes, evaluates, and compares the findings from previous studies that employ traditional RVS methods within a framework, and recommends the methods to be used for specific purposes and future enhancement techniques. This research guides the reader in comprehending the method they will choose on the basis of the site-specific building characteristics, as well as the types of adjustments and/or techniques (machine learning, fuzzy logic, neural networks, and simplified seismic assessment methods) that may be used to enhance the current conventional methods. For this reason, complementary information is offered, followed by research examples, including a comprehensive report on the sample applications in different countries. Additionally, the historical development of the RVS methods are presented from the beginning.

The study is structured in six sections, as follows:

- Section 2 examines the extensively used RVS methods developed over the last 30 years;
- Section 3 provides a brief description of the regionally utilized RVS procedures that have partially been developed on the basis of the techniques presented in Section 2, and it describes the RVS methodologies developed for special types of buildings;
- Section 4 introduces an overview of the research projects and compares the RVS techniques;
- Section 5 is devoted to a discussion of the RVS methodologies;
- Section 6 offers conclusions and the authors' recommendations for future studies for the further review of RVS techniques.

## 2. Review of Widely Used RVS Methods Developed over the Past 30 Years

According to Sinha and Goyal (2004), the rapid visual screening (RVS) technique is conducted by performing a street survey within a short period of time (15 to 30 min), and without performing structural calculations [51]. The RVS scoring methodology is used to determine the building performance score using a scoring system that consists of basic calculations [52]. As stated in FEMA P-154 [8], buildings that are hazardous in terms of earthquakes are determined by using a multistage screening procedure. At this stage, the features of the buildings that are visible from the outside, and that affect the building seismic performance, are visually inspected from the street. If needed, the necessary calculations are performed [53].

The initial scoring methodology was recommended in California in the mid-1970s, and was followed by approaches for evaluating the likelihood of damage on the basis of expert opinion in the mid-1980s [54]. Figure 2 illustrates the chronology of the widely used RVS methods that were designed following the first scoring methodology introduced in California. Later, several researchers proposed different RVS methodologies, which are the approximate seismic vulnerability assessment techniques. These methods can easily be applied to large-scale seismic risk management, and they are divided into two main categories:

(1) Techniques for using instrumental and quantitative data; and
(2) Methods for the acquisition of both quantitative and qualitative data [55].

RVS is an effective rapid method for obtaining data as a field-based building inventory, including the site assessment, the data collection phase, and the data analysis, in order to make structural assessment decisions [53]. RVS methodologies include the identification of the basic elements and their properties that make up the structure, the assessment of the structural redundancies, the regularity of the mass and the load direction, the vertical smoothness, the seismic improvement types, and the construction quality. The visual investigation of buildings consists of the general structural information, the structural system, the nonstructural system (wall, roof, etc.), and the foundation information. The general structural details include the year of completion, the location, and the quality of the structural appearance. The structural system is to be determined at the beginning of the inquiry. In order to explore the structural system details relevant to the RVS technique properly,

the building type is specified at the beginning of the site inspection. Following this, the current features of the buildings, presented in Table 1, are examined in order to identify the characteristics of the existing buildings.

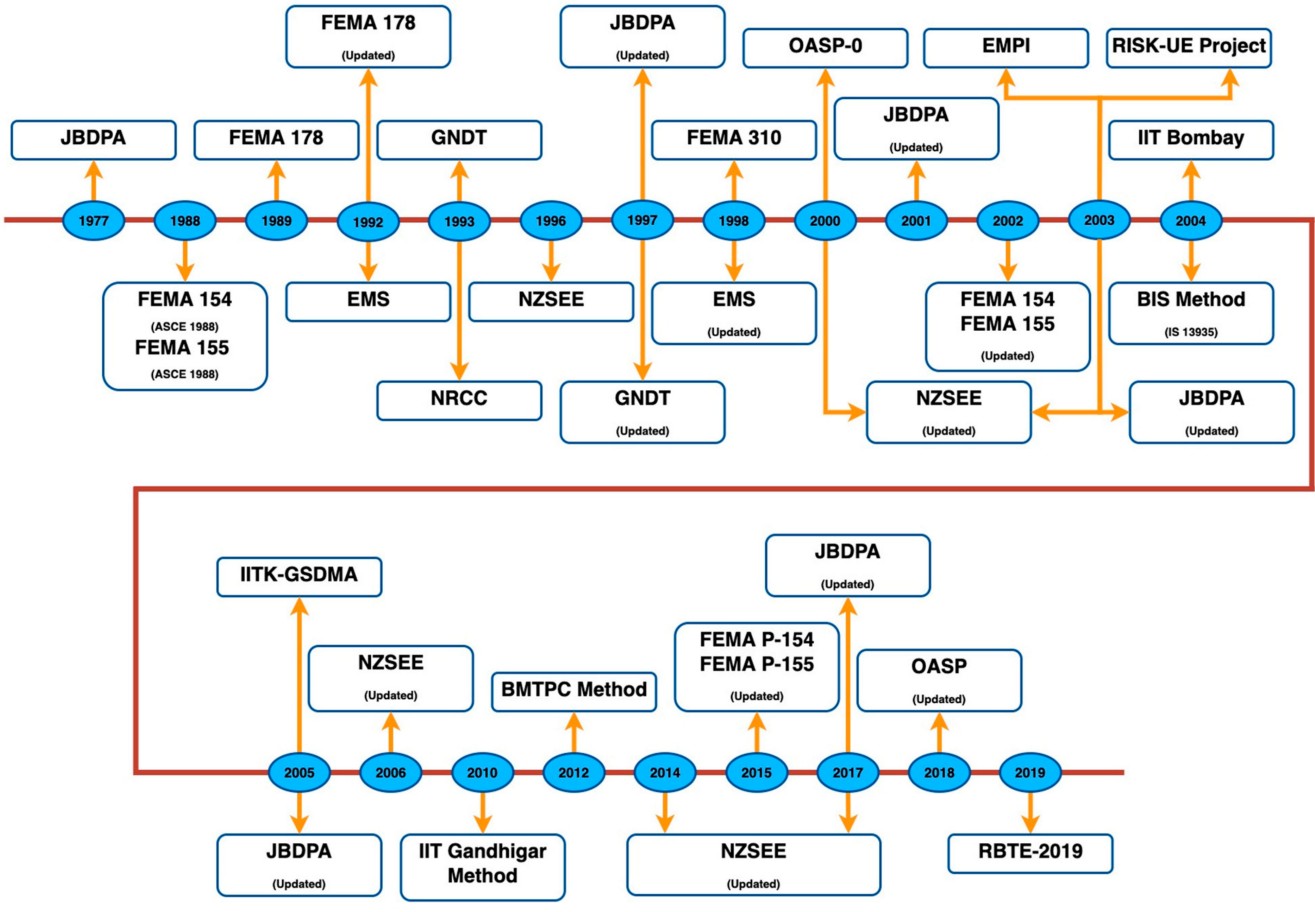

**Figure 2.** Timeline graphic of the development of the RVS methods.

**Table 1.** Some of the building characteristics used to examine existing buildings.

| | | |
|---|---|---|
| • Form of structure | • Number of floors | • Floor height |
| • Soft and weak stories | • Column/beam size | • Reinforcement types and dimensions |
| • Short/fixed column | • Steel section types | • Inconsistencies in plan |
| • Types of connection | • Timber type | • Inconsistencies in elevation |
| • Mortar deterioration | • Masonry deterioration | • Concrete deterioration |

Moreover, the structural and nonstructural systems affect the structural response under seismic excitation. The characteristics of the walls (see Table 2) are considered in order to evaluate the effect of the wall, as a nonstructural system element, on the overall structural safety [56]. The characteristics of the roof structure, shown in Table 2, affect the structural response under seismic excitation [56,57]. Parapets and other similar elements should also be evaluated in order to define the complete structural system properties [58].

**Table 2.** Wall and roof structure characteristics that affect the structural response.

| Wall | | Roof |
|---|---|---|
| • Wall type | • Wall connections | • Roof structure type |
| • Vertical bands | • Horizontal bands | • Roof shape |
| • Wall offset | • Filling material | • Roofing material |
| • Damage to the boundary wall | • Size and placement of openings | • The roof and structure connection |
| • Wall dimensions (thickness, length, height) | | |

RVS techniques can be based on a scoring system, the seismic index method, or the vulnerability index to calculate the expected damage of a structure, and to decide if the building requires a next-stage evaluation process with respect to the data collected, without performing any analytical structural seismic analyses. The application of the structural scoring system allows practitioners to identify the lateral-load-resisting systems of the buildings, and to assess the structural system on the basis of the previously determined fragility curves of the special building type. This allows for the identification of the building class and for an examination of the structural vulnerability. The scoring systems are based on the anticipated imminent earthquake intensity, the building stock properties, and the structural design standards in the selected region. The screening results may be used to evaluate the risk of partial or full collapse, as is explained by Lizundia et al. (2014) [59]. However, the screening findings are not used to define the safety level of a building, but are used to identify the "damageability of buildings" [60]. This technique saves time and money and it is easy to use for buildings that do not require detailed assessments [8,51,61–63].

Assessment errors can also be considered in the evaluation of the questionnaire forms. These errors arise when the buildings are surveyed by inexperienced engineers, or when the questionnaire is not detailed enough. This usually happens when sufficient training is not provided to minimize the surveyor bias. These types of errors restrict the implementation of the database-based functions in other locations. According to Tesfamariam and Liu [64], the effect of such errors on the results has not been clearly explained because the current studies do not evaluate the damage data independently.

Each of the conventional RVS methods, which were developed and have been extensively used over the past 30 years, are described in depth below.

*2.1. RVS Standards of the United States*

The RVS methods that were developed for the seismic assessment of buildings in the United States consist of FEMA 154 [8–10], FEMA 155 [65–67], and FEMA 310 [68]. These methods are described below.

2.1.1. FEMA 154

The FEMA released the first edition of the RVS standard for buildings in 1988 as the American Society of Civil Engineers (ASCE) "Rapid Visual Screening of Buildings for Potential Seismic Hazards: A Handbook" of the American Society of Civil Engineers (ASCE). The FEMA RVS scoring system was revised according to the latest technological developments, user feedback, the most recent information on seismic hazard mapping, the evaluations based on fragility curves, and the inferences taken from the 1990s earthquakes (1989 Loma Prieta; 1992 Landers; 1992 Big Bear; 1994 Northridge) as FEMA 154 [10] in 2002 [10,69,70]. The third edition of the RVS methodology, FEMA P-154 [8], was revised in 2015 and included not only information about the building-type definitions and key features, but also a completed screening form and RVS program management [15].

In the initial version of the FEMA 154 RVS technique [9], the seismic hazard is assessed using modified Mercalli intensity (MMI)-based damage probability matrices and California earthquake (1989 Loma Prieta; 1994 Northridge) damage assessment data for the basic structural hazard (BSH) scores [67]. The Hazards United States (HAZUS) fragility curves, along with the design maximum considered earthquake (MCE), are utilized to calculate

the BSH scores of the second edition FEMA 154 RVS technique [10]. By using the HAZUS fragility curves and analytical computations, the BSH scores are adjusted for the building types addressed in FEMA P-154 [8].

According to the most recent version of FEMA 154, the first step in the implementation of the RVS approach is examining the structural system and deciding on the classification of the buildings, out of the 17 structural classes, on the basis of the structural materials and systems. In comparison to the previous edition, two new building types have been included. The building properties are determined in order to evaluate the probable vulnerability by visual inspection. The building identification information consists of the building identification, the latitude and longitude, the site seismicity, a screener identification, the building characteristics, and photographs of the building. The building occupancy is based on the parameters shown in Table 3.

**Table 3.** Parameter-affected building occupancy.

| | | |
|---|---|---|
| • Occupancy classes | • Additional designations | • Soil type |
| • Geological hazards | • Plan irregularities | • Vertical irregularities |
| • Adjacency | • Exterior falling hazards | • Damage deterioration |
| • Comments section | | |

The score modifiers (SMs) are used to calculate the RVS of a structure by using a scoring matrix that depends on the FEMA building types. The considered features for evaluating the SMs are the vertical irregularity, the plan irregularity, the pre-code or post-benchmark design, the soil type (from A, meaning hard rock, through to F, meaning poor soil), and the minimum score ($S_{MIN}$). As a result of calculating the worst possible combinations, the $S_{MIN}$ can become zero, or even negative. In this context, the negative values show more than 100% damage. In order to overcome this problem, the $S_{MIN}$ value was defined on the data collection form. The Level 1 final score ($S_{L1}$ and $S_{L2}$) is determined on the basis of the SMs for the selected building, and the basic score is determined for a specific kind of structure on the screening form. The scores used in the scoring system of the RVS methods were revised by employing numerical equations and HAZUS fragility curves.

The FEMA P-154 RVS method includes two-level screening forms, i.e., Level 1 and Level 2, for each seismicity region, which can be moderate, moderately high, high, and very high. The Level 2 screening form, which has more SMs to evaluate the final score (FS), and which requires a more qualified screener, is more detailed and is optional. At the end of the Level 2 screening form, some of the nonstructural screening properties are considered. The FS ranges between 0 (potential for collapse) and 7 (better expected seismic performance). On the basis of the current seismic design requirements, 2 is recommended as the "cut-off" value of the FS. The RVS methodology divides buildings into two categories:

- Buildings having acceptable seismic performances ($2 < FS \leq 7$); and
- Buildings that are seismically hazardous and need to be assessed with further detailed methodologies by a structural design professional specializing in seismic design ($0 \leq FS \leq 2$).

### 2.1.2. FEMA 155

The first edition FEMA 155 [65], which is the supporting documentation for the RVS method that was prepared by the ATC for the FEMA, was released in 1988. The second edition of FEMA 155 [67], which is the second edition of the RVS supporting documentation, was published in 2002 and categorizes structures on the basis of their level of vulnerability. These vulnerability levels are as follows:

(1) The building poses a reasonable risk to life safety; and
(2) The building is seismically hazardous [71].

The third edition of FEMA P-155 [66], which is entitled, "Rapid visual screening of buildings for potential seismic hazards: Supporting documentation", was released in 2015

and was based on FEMA 155 (2002) [67] to evaluate the basic scores and the SMs. FEMA P-155 is written for those who want to understand the details and assumptions that underlie the methodology, and how the basic scores and SMs were calculated on the basis of the capacity spectrum and the fragility curves. The RVS method also explains how the risk depends on the scores, as well as on the effect of the pounding and building additions shown in Equation (1).

$$S = -\log_{10}(P[Collapse|MCE_R \text{ ground motions}]) \tag{1}$$

where S denotes the FS on the basis of the maximum considered earthquake (MCE$_R$). FEMA P-154 [8] is used to define the collapse, and later, FEMA P-155 [66] is utilized to evaluate the probability of collapse. In this term, the risk score (S$_R$) is based on HAZUS methodology:

$$S = -\log_{10}(P) \quad P = \left(\frac{1}{10}\right)^S \tag{2}$$

where S is equal to 1, and P = 0.1, or a 10% probability of collapse. For developing the fragility curves, a factored lognormal cumulative distribution function was used, as is presented in Equation (3):

$$P = P_c \cdot y(x) \quad y(x) = \Phi\left(\frac{\ln\left(\frac{x}{\theta}\right)}{\beta}\right) \tag{3}$$

where P$_c$ is the collapsed part of a building; y is the likelihood of total structural collapse; $\Phi$ is the factored lognormal cumulative distribution function; x is based on the F$_v$S$_1$ for different seismicity regions; $\theta$ denotes the mean; and $\beta$ denotes the logarithmic standard deviation given in Equation (4):

$$\beta = \frac{\ln\left(\frac{x_{0.10}}{\theta}\right)}{\Phi^{-1}(0.10)} = \frac{\ln\left(\frac{x_{0.10}}{\theta}\right)}{-1.28} \tag{4}$$

where the P$_c$, x$_{0.10}$, $\theta$, and $\beta$ values are based on the FEMA building types presented in FEMA P-155 (Table 8-1 of the code).

### 2.1.3. FEMA 310

FEMA 310 [68], which is entitled, "Handbook for the seismic evaluation of buildings: A prestandard", was developed in line with the revised FEMA 178 report (1992) [72] from the FEMA 178 report (1989) [73], "NEHRP Handbook for the Seismic Evaluation of Existing Buildings", which was published by the FEMA in 1998 for the seismic assessment of the existing buildings. "Prestandard" indicates that it was still in the development process. Many earthquakes occurred throughout the world during the development of the FEMA 178 report and after its publication. These earthquakes revealed additional information in some parts of the FEMA 178 report that needed to be modified. FEMA 310 was published in order to include technological advances, the lessons learned from recent earthquakes, and to incorporate the experiences of design professionals.

As is stated in FEMA 310 [68], it is recommended that a rapid visual screening be performed using FEMA 154 and 155 (1988) before implementing this document. While FEMA 310 contains new types of buildings, the considered seismic probability of the exceedance was modified from 10 to 2% within 50 years, and the immediate occupancy performance class was included, in addition to the life safety performance class [68,74,75]. However, contradictions in the method have been discovered:

- Despite the substantial damage in the structural and nonstructural components in the life safety performance class, the risk of total damage and the occurrence of life-threatening injuries are minimal [74,76];

- Under seismic loads, minimal damage can be found in the structural and nonstructural elements of the building during the immediate occupancy performance class, while total collapse does not occur [76–79];
- Although there could be minor cracks in the exterior walls of URM structures, as is shown in Figure 3, for the initial occupancy damage state, they have no effect on the performance of the load-bearing system under lateral loads.

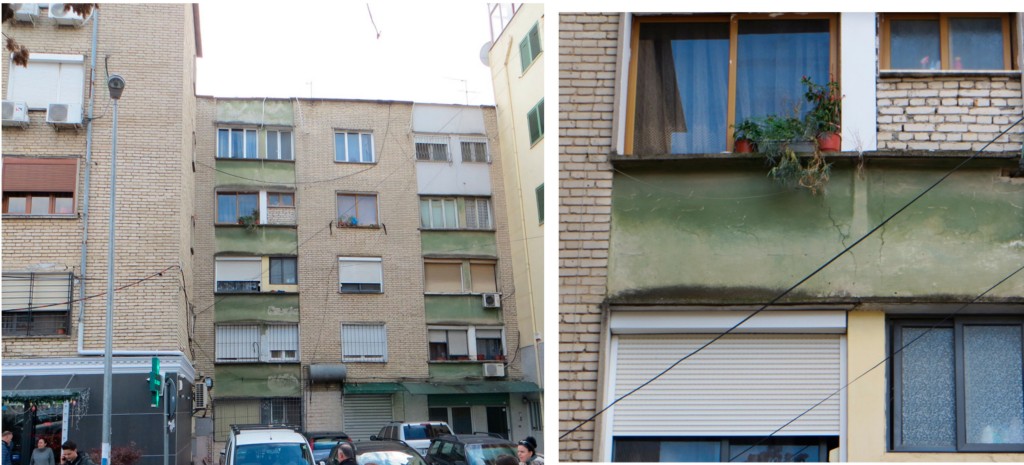

**Figure 3.** A sample initial occupancy damage state for URM buildings from 2019 Albania earthquake (taken by authors).

FEMA 310 is a three-stage seismic evaluation tool. The assessment stages from Tiers 1 to 3 are classified into three groups: a screening phase, an evaluation phase, and a comprehensive assessment phase, respectively. The Tier 1 (RVS) evaluation approach examines the structural, nonstructural, and foundation properties to fill the forms for the selected performance level on the basis of the seismicity of a region.

FEMA 310 considers 24 types of buildings, as compared to the 17 classes of FEMA P-154 [8]. The Tier 1 assessment process consists of three checklists that are used to examine the building deficiencies: the structural system; the foundation and the geotechnical hazards; and the nonstructural system checklists. A building is referred to as a "benchmark building" if it is constructed to satisfy the specifications of the required design legislation and if this is implemented properly [74]. If a structure is designated as a "benchmark building", the structural examination step can be ignored [75,78]. The need for a detailed evaluation is decided after the first stage of the assessment process. On the basis of the three basic seismicity zones, the significance of the cost–benefit relation, or other advantages, would be considered in order to make a further detailed evaluation decision [68,80]. In comparison to the previous FEMA method, FEMA 310 considers not only the basic structural and nonstructural checklists, but also the supplemental structural and nonstructural checklists for each building type. In cases where the seismicity level is moderate to high, a building is assessed using the immediate occupancy performance level supplementary checklist [74]. The checklists are selected on the basis of the anticipated level of performance (e.g., life safety and immediate occupancy) and the seismicity of the considered region (e.g., low, moderate, and high), as is shown in Table 4.

The checklists involve the structural components, the geologic site and foundation components, the nonstructural components, the building system, the lateral-force-resisting system, the diaphragms, the connections, the configurations, and the condition of the materials. The details of some of the parameters mentioned above are presented in Table 5.

When an existing building does not show the required capacity as a result of the first stage assessment, a detailed evaluation can be performed using the second and third stage assessments, respectively [74,81].

Although it was originally recommended that structures be inspected utilizing FEMA 154 [9] prior to performing FEMA 310, FEMA 154 was modified in 2002 [10] and 2015 [8]. In this case, comparing FEMA 310 with the modified FEMA 154 methods [8–10] will demonstrate the significance of the modifications.

**Table 4.** FEMA 310 Tier 1 checklist selection [68].

| Region of Seismicity | Level of Performance | Required Checklists | | | | | |
|---|---|---|---|---|---|---|---|
| | | Region of Low Seismicity | Basic Structural | Supplemental Structural | Geologic Site Hazard and Foundation | Basic Nonstructural | Supplemental Nonstructural |
| Low | LS | ✓ | | | | | |
| | IO | | ✓ | | ✓ | ✓ | |
| Moderate | LS | | ✓ | | ✓ | ✓ | |
| | IO | | ✓ | ✓ | ✓ | ✓ | ✓ |
| High | LS | | ✓ | ✓ | ✓ | ✓ | |
| | IO | | ✓ | ✓ | ✓ | ✓ | ✓ |

**Table 5.** Parameters presented in FEMA 310 checklists.

**Lateral-Force-Resisting System**

- Moment frames
- Concrete moment frames

- Shear walls
- Reinforced masonry shear walls
- Walls in wood-frame buildings

- Moment frames with infill walls
- Precast concrete moment frames

- Concrete shear walls
- Unreinforced masonry shear walls
- Braced frames

- Steel moment frames
- Frames not part of the lateral-force-resisting system

- Precast concrete shear walls
- Infill walls in frames
- Concentrically braced frames

**Diaphragms**

- Precast concrete
- Wood
- Metal deck
- Other

**Connections**

- Anchorage for normal forces
- Vertical components

- Shear transfer
- Panel connections

- Interconnection of elements

**Geologic Site Hazards and Foundation Checklist**

- Geologic Site Hazards
- Condition of Foundations

**Nonstructural Checklist**

- Partitions
- Cladding and glazing
- Masonry chimneys
- Mechanical and electrical equipment

- Metal stud back-up systems
- Concrete block and masonry back-up systems

- Ceiling systems
- Masonry veneer
- Stairs
- Parapets, cornices, ornamentation, and appendages

- Ducts

- Light fixtures
- Piping
- Building contents and furnishing
- Hazardous materials storage and distribution

- Elevator

### 2.2. RVS Methodologies Developed by the European Union

The RVS methodology for the seismic assessment of buildings in Europe consists of the EMS-98 [11] and the RISK-UE Project [12].

### 2.2.1. EMS-98 Scale

The first edition of the European macroseismic scale (EMS) was developed in 1992 by the European Seismological Commission (ESC) in Prague. The second edition of the EMS [11] was proposed in 1998 and is due to the occurrence of significant earthquakes both in Europe and around the world. Prior to the development of the EMS-98 scale, the Medvedev–Sponheuer–Karnik (MSK) scale was used to evaluate building types [11].

The EMS-98 scale was developed by reviewing and expanding the MSK-64 scale, and the later modified version, as the MSK-81 scale [11,82]. The MSK scale-based developed EMS-98 scale is employed to classify buildings both qualitatively (building type and vulnerability) and quantitatively (indicating degrees of damage) [11,82,83]. The damage probability matrixes (DPMs) obtained from the 1980 Irpinia earthquake were used to conduct the first rapid verification of the results obtained using the EMS-98 [84].

Masonry, reinforced concrete (RC), steel, and wood structures were basically considered by the EMS-98 [11] to be the most commonly used building construction systems in Europe. The masonry structures and the RC buildings can be distinguished as depicted in Table 6.

**Table 6.** Classification of masonry and reinforced concrete structures.

| | | |
|---|---|---|
| **Masonry Structures** | <ul><li>Simple stone</li><li>Massive stone</li><li>Reinforced brick</li><li>Confined masonry</li></ul> | <ul><li>Rubble stone/fieldstone</li><li>Massive stone</li><li>Unreinforced brick/concrete blocks</li><li>Unreinforced brick with RC floors</li></ul> |
| **RC Structures** | <ul><li>RC frame structures</li></ul> | <ul><li>RC wall structures</li></ul> |

This approach defines six vulnerability classes, from A to F, depending on the building typology. The damage classifications consist of five damage states, from Grades One to Five, in between the EMS-98 scales of V and XII, respectively:

- Negligible to slight damage (no structural damage, slight nonstructural damage);
- Moderate damage (slight structural damage, moderate nonstructural damage);
- Substantial to heavy damage (moderate structural damage, heavy nonstructural damage);
- Very heavy damage (heavy structural damage, very heavy nonstructural damage);
- Destruction (very heavy structural damage).

In addition, the damage is considered to indicate the safety of the building against earthquakes [85].

The uncertainties and inadequacies that may arise in the identified damage classes of buildings are included in the EMS-98 damage intensities [86]. The linguistic terms employed to characterize the various damage states (e.g., less, much, most, etc.) are significantly ambiguous [86–88]. Furthermore, uncertainty in the information precludes the application of the DPM [86] because the intensity is perceived as a result of the earthquake impact on humans; the intensity described on the basis of their effects on structures may differ; i.e., there could be inconsistencies in the data [11]. Researchers propose that the linguistic frequencies specified for the different damage classes and macroseismic intensity levels should be assessed using fuzzy sets [5,79,86,89].

The vulnerability model in the EMS-98 is incomplete and uncertain, and the uncertainty can be handled by using a fuzzy set theory together with probability theory [12,85]. Moreover, as Giovinazzi et al. [90] stated in the EMS-98 methodology, the fuzzy set theory can be used alongside with a classical probability theory to overcome the uncertainties. The macroseismic model can initially be calibrated on the basis of postearthquake data and, as proposed by Giovinazzi and Lagomarsino [91], on the basis of the EMS-98 [85]. Giovinazzi [92], Giovinazzi and Lagomarsino [93], and Lagomarsino and Giovinazzi [88] made further developments in the macroseismic model, which uses classical probability theory and fuzzy set theory in order to determine the vulnerability of buildings on the basis of the EMS-98 in European areas [11], as explained by Gueguen [85]. Ademović et al. [94] conducted an assessment on the seismic vulnerability of existing masonry buildings in Bosnia and Herzegovina in 2020 using the macroseismic method adapted from the EMS-98 scale and the vulnerability index method (VIM). In Cologne, Germany, the EMS-98 [11] was used to demonstrate five possible vulnerability classes of the existing buildings using DPMs, on the basis of five damage states or fragility curves [95,96].

In conclusion, while the EMS method could be used for postearthquake building screening because of its simplicity of use and its ability to define damage states on the basis of a short description, this method is insufficient for determining pre-earthquake damage states on the basis of our experience. When comparing the EMS approach to other pre-earthquake RVS methods (e.g., FEMA 154, FEMA 310), it becomes clear that pre-earthquake building inspection forms are needed for its use.

### 2.2.2. The RISK-UE Project

The RISK-UE project was funded to assess the seismic vulnerability of cities and regions in Europe by an advanced method by the European Commission in 1999, and it was supported by the European Union (EU). The damage estimation procedure of the RISK-UE [12] in 2003 followed three steps: the LM1, LM2, and LM3 methods.

The vulnerability model defined in the European macroseismic scale (EMS-98 [11]) serves as the basis for the macroseismic model (LM1) methodology, and it is derived from Giovinazzi and Lagomarsino [93,97] to examine the seismic response of a particular urban area [98]. Buildings are classified into 23 groups by the LM1, on the bases of the structural material and the building type, as is shown in Table 7 [99]. Moreover, each building type can be divided into three subclasses: low-rise, mid-rise, and high-rise. As a result, buildings can be classified into 65 classes on the basis of the performance-based building characteristics, and the major variables influencing the probable functionality loss and the degradation [100].

**Table 7.** The considered building classes in the macroseismic model (LM1).

| Building Classes | | |
|---|---|---|
| Rubble stone, field stone | Simple stone | Massive stone |
| Adobe | Wooden slabs (URM) | Masonry vaults (URM) |
| Composite slabs (URM) | RC slabs (URM) | Reinforced or confined masonry |
| Overall strengthened masonry | RC moment frames | RC shear walls |
| Regularly infilled RC frames | Irregular RC frames | RC dual systems |
| Precast concrete tilt-up walls | Steel frames with URM infill walls | Steel moment frames |
| Steel braced frames | Wooden structures | Steel and RC composite systems |
| Precast concrete frames with concrete shear walls | Steel frames with cast-in-place concrete shear walls | |

To create the RISK-UE building-classification-matrix-related DPMs, the LM1 approach is used to identify the vulnerability indices and the vulnerability classes and to determine them into three steps for the five damage grades ($D_k$) (k = 1, . . . , 5):

1. The vulnerability index ($V_I$) calculation of a single building:

$$V_I = V_I^* + \Delta V_R + \Delta V_m \quad \Delta V_m = \sum V_m \tag{5}$$

where $V_I$ is the vulnerability index; $V_I^*$ is the typological vulnerability index; ($\Delta V_m$) is the regional vulnerability factor, which is based on the building and the region characteristics given in Equation (5); and ($\Delta V_R$) is the seismic behavior modifier, which is based on previous damage data or expert judgement [101];

2. The calculation of the mean damage grade ($\mu_D$). The mean damage grade ($\mu_D$) estimation is given in Equation (6):

$$\mu_D = 2.5 \left[ 1 + \tanh\left( \frac{I + 6.25 \cdot V_I - 13.1}{Q} \right) \right] \tag{6}$$

where Q is the ductility index that is based on the typology of the building and the construction features; and I is the macroseismic (EMS-98) intensity, which usually varies between V and XII, and is expressed in Roman numerals. For residential buildings, 2.3 is selected as the Q;

3. Damage distribution evaluation (DPMs and fragility curve estimation).

On the basis of the probability mass function of the binomial distribution, the probability of having damage, $P(D_k)$, which is a statistical demonstration of DPMs, is calculated with Equation (7) [94,102,103]. For various damage grades, the fragility curves consist of the y − axis ($P[D_k]$), which is shown in Equation (8), and the x − axis ($I_{EMS-98}$) [102]:

$$P_\beta(k) = \frac{5!}{k! \cdot (5-k)!} \cdot \left(\frac{\mu_D}{5}\right)^k \cdot \left(1 - \frac{\mu_D}{5}\right)^{5-k} \tag{7}$$

$$P(D \geq D_k) = 1 - P_\beta(k) \tag{8}$$

where k = 0 to 5 is based on the EMS-98 damage thresholds; ! is the factorial sign; and $\mu_D$ is the weighted mean damage value. The mean damage grade ($\mu_D$) estimation is given in Equation (9), which has values ranging between 0 and 5, which are represented as $D_k$. The mean damage grade estimation represents the estimated discrete damage distribution on the basis of the five damage grades described in the EMS-98 [94]:

$$\mu_D = \sum_{k=0}^{5} P_\beta(k) * D_k \tag{9}$$

where $P_\beta(k)$ are the discrete probability values for each damage grade shown in Equation (7). The macroseismic model (LM1) of the RISK-UE has been used by Boutaraa et al. [99] to assess the buildings of Chlef City in Algeria.

The microseismic model (LM2) of the RISK-UE, which is based on the capacity and fragility model, has been used to examine urban seismic risk [90,98]. By using capacity curves (pushover curves), Figure 4 is obtained, which depicts the nonlinear behavior of the models and the variations between the design, the yield, and the ultimate structural strength level points. Fragility curves are used to estimate the likelihood of being in, or exceeding, a certain damage scenario, and they are explained in detail by Milutinovic and Trendafiloski [12]. Boutaraa et al. [99] indicate that the RISK-UE LM2 approach is appropriate for particular locations with comprehensive seismic site data, depending on the spectral acceleration, the velocity, and the displacement values.

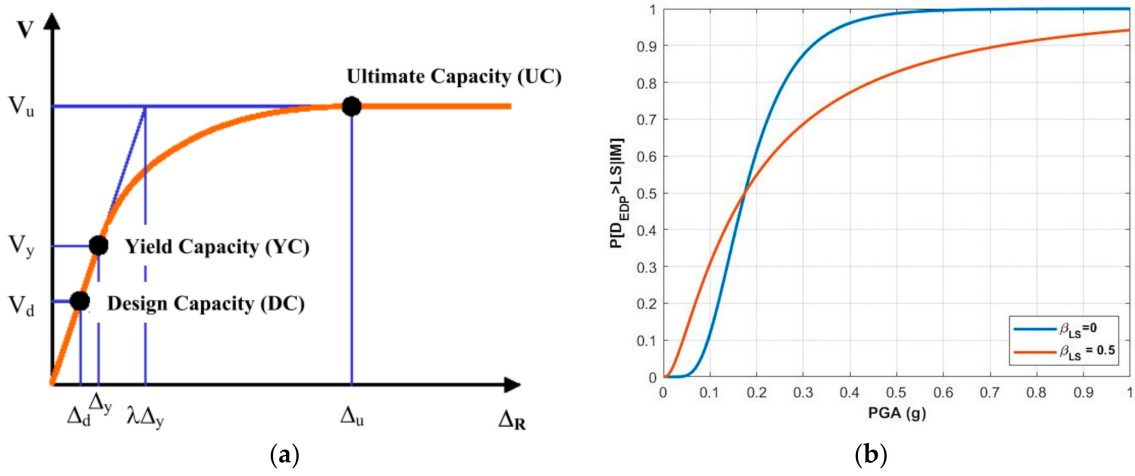

**Figure 4.** (**a**) Building capacity curve, and (**b**) fragility model.

A comparison of the damage classes described in the LM1 and LM2 methods shows that the LM1 method consists of five types of damage, and that there is no visible damage (none). However, the LM2 consists of four types of damage and none as a classifier of the damage state. In the EMS-98-based LM1 method and the FEMA/NIBS-based LM2 method, the damage grades between D0 and D2 correspond to each other. However, the D4 and D5 damage classes in the LM1 method are characterized as very heavy and destructive,

respectively, but the D4 damage class in the LM2 method is expressed as complete, as is shown in Table 8.

**Table 8.** LM1, LM2, and HAZUS damage grade relations (Compiled based on [12]).

| Damage Grade | | | Damage Grade Label | | |
| --- | --- | --- | --- | --- | --- |
| | | | LM1 | LM2 | FEMA/NIBS (HAZUS) |
| 0 | - | D0 | None | None | None |
| 1 | - | D1 | Slight | Minor | Slight |
| 2 | - | D2 | Moderate | Moderate | Moderate |
| 3 | - | D3 | Substantial to heavy | Severe | Extensive |
| 4 | - | D4 | Very heavy | Complete | Complete |
| 5 | - | D5 | Destruction | | |

In the LM3 Method, the structural seismic response is calculated through a nonlinear dynamic analysis (NLDA) by using comprehensive acceleration time histories (earthquake records) [98].

*2.3. RVS Methodology Applied in Japan*

Tokachi-oki, which struck in 1968, was the first large earthquake in Japan to cause severe structural damage to reinforced concrete structures [104–106]. In 1976, a commission was organized by the Ministry of Construction to develop a method for determining the earthquake resilience of the existing buildings [104]. The JBDPA's index method consists of three stages [105,107]. The first stage assessment procedure for RC buildings was introduced in 1977 by the Japan Building Disaster Prevention Association (JBDPA). This evaluation procedure requires the material properties and the structural element dimensions of the considered buildings [104,105,108]. The Japanese RVS methodology was developed by the JBDPA [109–113] as the "Standard for Seismic Evaluation of Existing Reinforced Concrete Buildings". The design of the structures was based on the Japanese standard needs to define columns and shear walls with large dimensions, compared to the other design codes [17]. Therefore, the Japanese building screening technique is based on each story's shear capacity estimation, which involves the capacities of the columns and the shear walls [114]. The JBDPA screening method takes five times longer than the FEMA RVS method [108].

The nonstructural and structural responses are also affected by the consideration of the effect of the infill walls as nonstructural components. Infill walls, in general, improve the in-plane strength of the structural frame system [115]. The JBDPA consists of a three-stage screening procedure [114], and it considers both the structural and nonstructural elements on the basis of the seismic index of the structures ($I_S$). The $I_S$ consists of the basic seismic index of the structure ($E_0$) and other parameters, such as: the story-shear modification factor; the cumulative strength index ($C_T$); the irregularity index ($S_D$); the time index (T) given in Equation (10); and the seismic index of the nonstructural elements ($I_N$). The investigations that need to be conducted at the first-level inspection are the material properties, the cross-sectional dimensions, the structural deformations and concrete cracking, and the structural layout for the irregularity index. The first-level inspection is based on the estimation of the global shear strength [107]. The second-level screening considers the material properties, the cross-sectional dimensions, the structural deformations, the concrete cracking occurrences and ranges, the deterioration, and the aging grades and ranges. The second-level inspection consists of estimating the ultimate limit strength of the vertical load-carrying elements (columns and walls) [105,107]. The third stage consists of the detailed material properties handled by the sampling tests and dimensions, and of considering the cracks and defects to calculate the capacities of the structural members. The third-level inspection considers the estimation of the ultimate limit strength of the

load-carrying elements (columns, beams, and walls) on the basis of the structural drawings and a field examination [105,107]:

$$I_S = E_0 \cdot S_D \cdot T \tag{10}$$

$E_0$, which is the basic seismic index of the structure, is built by different equations for different screening levels. Following the $I_S$ value determination, a comparison with the $I_S$ is made using the seismic demand index ($I_{S0}$) formulation in Equation (11) to decide if the building has adequate strength under earthquake forces [116]. The calculations of the $I_S$ from Equation (10) are required for the two directions of each floor [107]:

$$I_{S0} = E_S \cdot Z \cdot G \cdot U \tag{11}$$

where the basic structural demand index ($E_S$) is evaluated on the basis of the 1968 Tokachi-oki and 1978 Miyagiken-oki earthquakes; the zone index (Z) considers the seismicity of the region; the ground index (G) considers the soil structure interaction and the geological conditions; and the usage index (U) considers the occupancy [107]. When comparing the $I_{S0}$ and $I_S$, if the $I_S$ is greater than the $I_{S0}$, the building has a low vulnerability condition. Although a low vulnerability condition indicates that there will be no large-scale structural damage, it does not mean that the structure will not be damaged. If the $I_S$ is smaller than the $I_{S0}$, there is the possibility that the building has a high vulnerability.

The abovementioned JDPA screening method appears to consider more comprehensive building information than the methods outlined in the preceding sections. Therefore, the JDPA technique may be adapted for different structure types (e.g., masonry), and compared to the previously stated methods, it demonstrates accuracy and applicability.

*2.4. RVS Methodology Used in New Zealand*

The initial RVS methodology of New Zealand was developed on the basis of the FEMA (1988) [9] in 1996, and it follows the procedure illustrated by the FEMA (1988) [57,80,117]. This code initially describes the assessment steps of existing buildings by an RVS method, which have varying structural configurations and material characteristics that were constructed before 1975 [78,118]. In terms of the types of structures considered and the score modifiers, the method of the New Zealand Society for Earthquake Engineering (NZSEE) differs from the FEMA method [118]. The procedures and the basic steps required for the seismic assessment of existing buildings consisting of different materials and structural configurations have been explained by the NZSEE (2000 [119]; 2003 [120]; 2006 [121]; 2014 [122]; 2017 [123]). The NZSEE recommends a two-stage seismic risk assessment procedure: an initial assessment procedure (IAP), which is the rapid evaluation stage; and a detailed seismic assessment (DSA), which consists of two parts: one for potential earthquake-prone buildings (EPBs) and one for non-EPBs only. The IAP procedure presents an initial evaluation phase of the existing buildings on the basis of the new building design regulations. The interaction curve, which is determined on the bases of the building gross area parameter and the final structural score, is used to decide whether a DSA is necessary [61,78,117]. The occupants of the building, as well as the causes of the structural damage, are reflected in the building gross area parameter [78,117].

EPBs provide engineers with a technical framework for the seismic evaluation of existing structures. This consists of the IAP for calculating the building seismic performance, and a brief explanation of the seismic risk assessment process. The new building standard (NBS) is derived on the basis of the characteristic features, such as the design year, the previous retrofitting interventions, the building importance, the fault distance, the site soil properties, and the vertical and plan irregularities integrated into the building area [78,117,124–126]. The earthquake rating evaluation is given in Equation (12):

$$\%NBS = \frac{Ultimate\ capacity\ (seismic)}{ULS\ seismic\ demand} * 100 \tag{12}$$

A value higher than 100% NBS can be obtained if a new building is designed and constructed under the requirements stated in the prescribed specifications and then examined according to this guideline. The ultimate capacity (seismic) is based on the primary lateral-load-carrying system, the site soil conditions, the probable capacities of the elements, etc. The ULS seismic demand is considered for an appropriate value of the structural performance factor ($S_p$). Table 9 shows the NBS grading system and the risk relation that is based on the new building and the life safety performance stage. On the basis of the %NBS grade, if %NBS $\leq$ 33, this indicates that the building is susceptible to a possible earthquake, and that the seismic vulnerability of the building should be calculated by a detailed assessment methodology. If 33 < %NBS < 67, then the structural system should be examined in a method more sophisticated than RVS. If 67 $\leq$ %NBS, then this indicates that the building could withstand imminent earthquakes.

**Table 9.** Potential building statuses based on seismic risk assessment of NZSEE (2017) (Compiled based on [127]).

| %*NBS* | Risk Level |
|---|---|
| >100 | Low |
| 80–100 | Low |
| 67–79 | Low to Medium |
| 34–66 | Medium |
| 20–34 | High |
| <20 | Very High |

The DSA is used to calculate the safety of a building against earthquakes, or to verify the accuracy of the results obtained with the IPA. It is also used to assess retrofitting needs on the basis of the DSA, and to evaluate the planned implementation strategy. Furthermore, the NZSEE differs from FEMA 154 in terms of its consideration to the URM cross-sectional area, the building plan area, the span-to-depth ratio, and other factors used when determining the attribute score.

*2.5. RVS Technique in Greece*

The Greek RVS methodology was developed by the Earthquake Planning and Protection Organization (OASP) as (OASP-0) [13] in 2000 on the basis of FEMA 154 (ATC-21) [9]. In this method, while an RVS method is used, the structural system, which consists of the structural elements, is also taken into account in the calculations, as well as the structural material properties. The soil category, the earthquake zone, the building type and characteristics, the soft-story existence, the short columns, the layout irregularities, the previous damage, the exterior condition of the building, and the construction year are used in order to consider the buildings with different characteristics found in Greece. For the classification of the buildings, 18 structural classification types were determined and were defined according to the initial structural hazard score (ISHS). Then, the provided ISHS is modified by determining the seismic zone, as well as the weak story and the regular arrangement of the masonry and the short column, to evaluate the basic structural hazard score (BSHS) [117,118]. As a result, the final score is determined from the BSHS, with some modifications. A final score of 2.0 and below should be examined in detail, as is suggested in the FEMA RVS method.

In addition, two scoring systems were proposed for a detailed evaluation of the hazardous buildings: (1) The OASP-R technique, which is defined on the basis of the OASP-0 [13]; and (2) The FEMA-G technique, which was developed on the basis of FEMA 154 [10]. Moreover, a fuzzy logic-based RVS methodology was developed by Demartinos and Dritsos (2006) [128] for the structural capacity categorization on the basis of five damage states in Greece [129]. For the secondary pre-earthquake control of the individual masonry

and RC buildings, two reports [130,131] were published by the Ministry of Infrastructure and Transport of Greece in 2018.

*2.6. Canadian RVS Methodology*

The National Research Council of Canada (NRCC) adopted a pre- and postdisaster vulnerability assessment mitigation technique for buildings in 1993 called the NRCC [14] "Manual for Screening of Buildings for Seismic Investigation," which mainly relies on the basis of FEMA 154 (ATC-21) [9]. The hazards to the structural components (e.g., beams, columns, foundation, slabs, etc.), the nonstructural components (e.g., partition walls, ceilings, etc.), and the building importance are interpreted by evaluating both the structural and nonstructural elements in the current NRCC [14] guidelines. A final "cut-off" score was developed, as was the case in FEMA 154 [80].

The key considerations of this screening technique are: the location and the age of the construction; the type and importance of the structure; the soil properties and the hazard zone; the falling hazard; the configuration of the irregularities in the plan and building elevation; and the nonstructural deficiencies. The NRCC [14] methodology relies on the seismic priority index (SPI), which comprises the summation of the structural index (SI) and the nonstructural index (NSI), as can be seen in Equation (16) [116]. The SI is based on the seismicity index, the soil properties, the structural category, the building irregularities, and the building importance. The NSI considers the soil properties, the building importance, and the maximum hazard value of life-threatening and critical buildings, as is stated by Kassem et al. (2020) [116].

$$
\begin{aligned}
SI &= A \cdot B \cdot C \cdot D \cdot E \\
NSI &= B \cdot E \cdot F \\
SPI &= SI + NSI
\end{aligned}
\tag{13}
$$

The following parameters are required for determining the SPI index, which is represented in Equation (13).

| | | | | | | |
|---|---|---|---|---|---|---|
| * A: | Seismicity | * B: | Soil conditions | * C: | Type of structure | |
| * D: | Irregularities | * E: | Building importance | * F: | Max ($F_1$, $F_2$) | * $F_1$: Falling hazards to life <br> * $F_2$: Hazards to vital operations |

A building constructed in compliance with the National Building Code of Canada (NBCC) is intended to have an SPI index score of 2.0 [132]. The score obtained by using the survey classifies the seismic vulnerability classes of the buildings as "low", "medium", and "high" seismic assessment stages [126]. If the SPI value of the evaluated building is less than 10, it is classified as "low"; if it is in the range from 10 to 20, it is classified as "medium"; and if the value is more than 20, then it is classified as being of the "high" seismic vulnerability class in terms of the priority of a further detailed assessment [116,132,133].

When comparing the NRCC and FEMA methodologies in terms of the parameters affecting the final score value, the building importance, the falling hazards to life, and the hazards to vital operations are considered to be parameters that are different than the parameters in the FEMA method in the NRCC. In terms of the required time for a field inspection of a single structure, the NRCC RVS method takes around one hour [14], whereas the FEMA method requires from 15 to 30 min [8]. Even though the NRCC RVS method takes more time than the FEMA RVS method, the NRCC RVS method seems to be more accurate since it considers more of the parameters that affect the structural performance [134]. By applying these slightly different methods to the same building stock after comparison to a DVA or postearthquake, the data could reveal the accuracies of the methods. Correspondingly, it is easy to adjust by considering the acquired comparison results because the FEMA method was developed using the capacity spectrum technique, whereas the NRCC method is difficult to modify as it was developed on the basis of an expert opinion.

### 2.7. RVS Methodology Developed in Italy

In 1997, the National Group for Earthquake Protection (Gruppo Nazionale per la Difesa dai Terremoti (GNDT)) developed the Post-Earthquake Damage and Safety Assessment (AeDES) protocol to create a structural risk assessment index that considers the numerous structural and nonstructural components. This approach uses a risk vulnerability index that is elaborated from the individual damage cases that are applied to various building components and weighted according to their relative extensions. Since GNDT methods are utilized to assess the postearthquake building operability, there is no regionally developed RVS approach in Italy for estimating the pre-earthquake building vulnerability, as is explained by Sangiorgio et al. [135].

The GNDT approach is divided into two submethodologies: (1) GNDT Level I; and (2) GNDT Level II [92]. Level I entails the gathering of the relevant information about the buildings to be examined. Level II is used to assess the fragility index of each building and is based on Benedetti and Petrini [136], Benedetti et al. [137], the GNDT [138], and Terremoti [139].

#### 2.7.1. Level I

The GNDT Level I method was established by the GNDT on the basis of earthquake statistical data processing from the November 1980 Irpinia earthquake, and later it considered other earthquakes as well [92,102]. The significant aspect of the seismic risk assessment is that it is concerned with the contributions of the various parameters, with different weights for the structural systems. Field surveys examine each parameter that has a fundamental effect on the structural vulnerability estimation, which is based on expert judgment and decision. DPMs indicate the probability of occurrence of a certain damage state in the considered building. Initially, DPMs were utilized to assess the damage states of buildings following the 1971 San Fernando earthquake [92]. In the GNDT Level I methodology, the DPM is created by involving three vulnerability classes: A, B, and C [102,116]. DPMs have been used in the GNDT and were designed with consideration to five damage state components and the Mercalli–Cancani–Sieberg (MCS) intensity [92].

#### 2.7.2. Level II

GNDT Level II is an empirical approach that employs a vulnerability index to quantify the degree of damage that might occur in a building [140,141]. The GNDT Level II method utilizes expert opinion and judgement on the basis of eleven weighted parameters, which are presented for the masonry and RC buildings in Table 10 [5,102,142]. However, wooden and steel buildings are not examined in the GNDT [143].

The vulnerability index ($I_v^*$) for GNDT II is based on the weighted sum of 11 building parameters and is calculated on the basis of the identified structural deficiencies in the structural system, as is shown in Equation (14):

$$I_v^* = \sum_{i=1}^{n} C_{vi} \cdot P_i \tag{14}$$

where n is the number of factors that are considered for the evaluation of the vulnerability index ($I_v^*$). The $C_{vi}$ values for the masonry and RC buildings, from the lowest to the highest vulnerability classes (A, B, C, and D), are shown in Table 10, respectively. $C_{vi}$ refers to the values assigned (0, 5, 25, and 45) for the masonry buildings, and to the variable values assigned between 0.25 and −2.45 for the RC buildings. $P_i$ denotes the weight assigned to each parameter on the basis of expert opinion.

**Table 10.** Value range of GNDT II vulnerability index for masonry and RC structures (Compiled based on [143,144]).

| Parameters | | $C_{vi}$ Interval | |
|---|---|---|---|
| | | **M** | **RC** |
| P1 | Type and organization of the resisting system | [0, 45] | [0, −2] |
| P2 | Quality of the resisting system | [0, 45] | [0, −0.50] |
| P3 | Conventional strength | [0, 45] | [0.25, −0.25] |
| P4 | Building position and foundations | [0, 45] | [0, −0.5] |
| P5 | Horizontal diaphragms | [0, 45] | [0, −0.50] |
| P6 | Plan configuration | [0, 45] | [0, −0.50] |
| P7 | Configuration in elevation | [0, 45] | [0, −1.50] |
| P8 | Maximum distance between walls | [0, 45] | [0, −0.50] |
| P9 | Roof | [0, 45] | [0, −0.50] |
| P10 | Nonstructural elements | [0, 45] | [0, −0.50] |
| P11 | Current condition | [0, 45] | [0, −2.45] |

RC: reinforced concrete structures; M: masonry structures.

According to Azizi-Bondarabadi et al. (2016) [102] and Giovinazzi (2005) [92], the normalized vulnerability index ($I_v$) is the normalization of the $I_v^*$ to classify the building damage under an impending earthquake. The formulation for the $I_v$ is presented in Equation (15):

$$I_v = \frac{I_v^*}{382.5} \tag{15}$$

The value, 382.5, is used to normalize the vulnerability index to yield a normalized vulnerability index for the masonry buildings that varies from 0 (least vulnerable) to 100 (worst case). In the case of the RC buildings, it ranges from −25 to 100. The mathematically linear relationship is established by utilizing a fragility index between the $y_i$ and $y_c$ values given by Equation (16):

$$\begin{aligned} y_i &= a_i \cdot \exp(-\beta_i \cdot v) \\ y_c &= [a_c + \beta_c \cdot v^\gamma]^{-1} \end{aligned} \tag{16}$$

The parameters, $a_i$, $\beta_i$, $a_c$, $\beta_c$, and $\gamma$, are determined on the basis of the least-squares minimization by Grimaz et al. (1996) [145]. While $y_i$ indicates minor damage with an initial cracking stage, $y_c$ indicates severe and widespread damage and that the building is close to collapse.

The GNDT II approach assesses the link between the seismic impact and the damage level on the basis of the vulnerability function. The link between the peak ground acceleration (PGA) and the density is given in Equation (17) [145,146]:

$$\ln(PGA) = a \cdot I_{MCS} - b \tag{17}$$

where a is equal to 0.602; b is given as 7.073; and $I_{MCS}$ is the MSC-scale-based intensity [145].

The normalization for the vulnerability index of the RC buildings is given in Equation (18). The vulnerability index of the RC structures was converted into the vulnerability index of the masonry buildings by using Equation (18), which ensures the opportunity to compare the masonry buildings to the RC buildings [116]:

$$\begin{aligned} I_v &= -10.0.7 I_v^* + 2.5175 \quad for \quad I_v^* > -6.5 \\ I_v &= -1.731 I_v^* + 2.5175 \quad for \quad I_v^* < -6.5 \end{aligned} \tag{18}$$

The study conducted by Cara (2016) [145] consists of a correlation of the EMS-98 and GNDT II to determine the damage grades. A comparison of the GNDT II approach with the EMS-98 could be implemented by converting the PGA value computed by Equation (17) into $I_{EMS-98}$ and transforming the economic damage index to the mean damage grade.

Another method that was developed on the basis of the GNDT II (1993) [147] methodology is the Macroseismic GNDT (M.GNDT) II, which is used for masonry buildings [148].

A comparison of the GNDT II and M.GNDT II approaches in terms of the weight factors is presented in Table 11. In order to adapt the GNDT II to the Algerian building stock, the M.GNDT II method was developed by Athmani et al. [141] in 2015 on the basis of GNDT II. Table 11 shows the parameters that were used to evaluate the vulnerability index.

**Table 11.** Comparison of vulnerability index parameters (Compiled based on [141]).

|  |  | Weight | |
| --- | --- | --- | --- |
|  | Parameter | **GNDT** | **M.GNDT** |
| P1 | Typology of resisting system | - | 2.50 |
| P2 | Organization of the resisting system | 1.00 | 1.00 |
| P3 | Conventional strength | 1.50 | 1.50 |
| P4 | Maximum distance between walls | 0.25 | 0.25 |
| P5 | Horizontal diaphragms | 1.00 | 1.00 |
| P6 | Number of floors | - | 0.75 |
| P7 | Location and soil conditions | 0.75 | 0.75 |
| P8 | Aggregate position and interaction | - | 0.75 |
| P9 | Plan regularity | 0.50 | 0.50 |
| P10 | Vertical regularity | 0.50 | 0.50 |
| P11 | Roof system | 0.25 | 0.25 |
| P12 | Interventions | - | 0.50 |
| P13 | General state of preservation | 1.00 | 1.00 |
| P14 | Nonstructural elements | 0.25 | 0.25 |

The vulnerability index ($I_v^*$) for the M.GNDT II is evaluated using Equation (14). The only difference in the equation is the number of parameters equal to 14, as is shown in Table 11.

The ignorance of the soil type in the GNDT method is the most indicative limitation, as is stated by Karbassi [77]. Apart from this, GNDT Levels I and II differ in terms of the damage definitions. GNDT Level I defines the damage by considering the macroseismic intensity, while GNDT Level II defines the hazard on the basis of the PGA [92].

For the field inspection of a single structure, the FEMA RVS approach takes 15 to 30 min [8], the NRCC RVS method takes 1 h [14], and the GNDT method takes from 1 to 1.5 h if the inside is accessible [143]. The additional data collection and analysis for the implementation of the GNDT approach can be completed in one to two days [143].

Furthermore, the GNDT method, created for postearthquake building assessments, needs to be modified for pre-earthquake screening in order to be compared to other methods in terms of the damage prediction capabilities. Aside from the RC and masonry structures, additional building types, such as wooden and steel structures, could be considered.

### 2.8. RVS Methodologies Developed for the Indian Building Stock

The materials of Indian rural and urban building stock consist of mud, earth, straw, wood, brick, stone, concrete, and steel [149]. Because of the distinct characteristics of the Indian building stock, a new RVS methodology was designed by the Indian Institute of Technology Kanpur (IITK), which was based on FEMA 154 [9,10], with the cooperation of the Gujarat State Disaster Mitigation Authority (GSDMA) with the "IITK-GSDMA Guidelines for Seismic Evaluation and Strengthening of Existing Buildings" (IITK-GSDMA) [150]. The adjustments made for the Indian conditions include both the parameters and the score values, as well as the calculation procedure, which is similar to FEMA 154 [8–10] by Sinha and Goyal [51]. The Level 1 evaluation step (RVS procedure) is the first-stage evaluation step and it considers both the RVS and the structural score evaluation for ten different Indian building types [151]. Four seismic zones are considered: low, moderate, high, and very high. For the suitable modifications, the structure type, the soil type (three soil types), and the related modifier values were changed. IS 13935 [152] was first suggested in 2004 for the screening of masonry structures, and it was revised as IS 13935 [153] in 2009 [154–157]. As defined by Sinha and Goyal (2004) [51], and Kapetana and Dritsos

(2007) [61], a final score value less than 0.7 shows the need for a further detailed assessment or retrofitting of the considered building.

During the data collection phase, the forms contain the basic score, the mid-rise and high-rise, the plan and vertical irregularities, the code detailing, and the soil type. The basic score is defined on the basis of the moment-resisting frame system [158]. The intended use of negative modifiers in the Indian RVS method indicates the subtraction of values from the basic score. A set of new parameters are considered, such as the number of stories, the re-entrant corners, the appearance and quality of the maintenance, open stories, short columns, and the presence of a basement, as explained by Sangiorgio et al. (2020) [135]. As Alam et al. (2012) [80] claim, specific types of buildings (e.g., nonductile RC frame and unreinforced masonry (URM) structures) have been given high priority for evaluation under this approach.

The IIT Bombay method, which was developed in 2004 by Sinha and Goyal [51], entails statistical methods. These statistical methods address the properties of the primary structural system [159] and the four building classes (reinforced concrete, steel, masonry, and timber) [70] in their calculations. In addition, the Indian standard BIS method, which was published in 2004 as, IS 13935, includes the RVS examination method for reinforced concrete and masonry structures, according to the Indian conditions [160]. The structural vulnerability calculations are made in accordance with the BIS method, depending on the capacity of the lateral-load-carrying system of the selected building. Another method is the IIT Gandhinagar method, which was modified by Jain et al. [151] in 2010 for the RVS of reinforced concrete structures on the basis of the Earthquake Master Plan of Istanbul (EMPI) [161], according to the Indian conditions. The Building Materials and Technology Promotion Council (BMTPC) method of Murty et al. (2012) [162] was developed by taking into account the material properties, the soil and ground properties, the structural system details, and the maintenance quality of buildings in India in 2012. This methodology includes the performance rating (PR) and the seismic safety index (SSI), which consider all of the abovementioned parameters [160]. Bhalkikar et al. (2021) [163] state that this is the most appropriate method in a multicriterion decision analysis, as it includes the most important observations of each criterion.

*2.9. RVS in Turkey*

Two methods, the EMPI [161] and the RBTE-2019 method [164], were developed for the rapid seismic risk assessment of buildings in Turkey.

2.9.1. EMPI Method

The Earthquake Master Plan of Istanbul (EMPI) [161] was developed in 2003 through the cooperation of two expert groups from four Turkish universities: Boğaziçi University; Yıldız Technical University; Istanbul Technical University; and Middle East Technical University. They considered existing reinforced concrete structures in the evaluation [165]. The EMPI [161] consists of a three-stage structural assessment, as is shown in Figure 5. The first stage is the RVS procedure, which is evaluated by using a sheet based on the seismic zone and the number of stories (less than eight-story RC buildings). The second stage includes a further detailed investigation of the structure by entering into the building and collecting the structural and nonstructural element data. The third stage is based on a highly sophisticated assessment of the considered building by using linear and nonlinear structural analyses, which require technical drawings of the structure.

The initial (basic) score ($B_S$) parameters are the number of stories and the local soil conditions, which affect the considered intensity of the ground motion in terms of the peak ground velocity (PGV) divided to three zones, as is shown in Table 12.

The vulnerability parameters ($V_P$) are the score modifiers. Their evaluation is shown in Table 13 and is based on the soft story, the heavy overhangs, the apparent building quality, the short columns, and the pounding effect, which can occur when the space within

the adjacent buildings is not adequate. Finally, the topographic effect is also regarded as the transfer of the seismic forces to the ground.

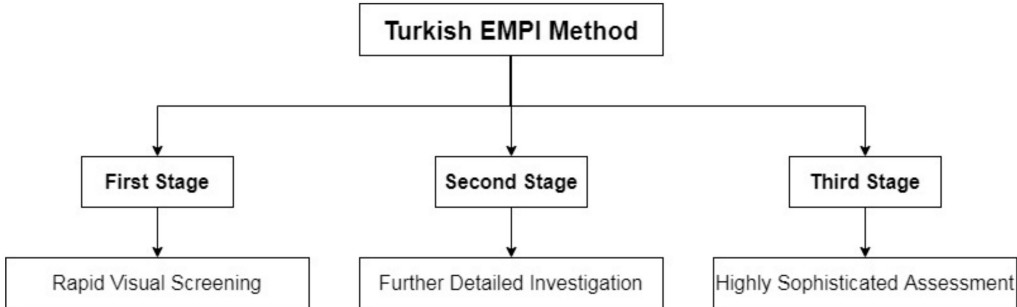

**Figure 5.** Flowchart of the Turkish RVS method (EMPI).

**Table 12.** EMPI seismic zone intensity classification (Compiled based on [161]).

| Seismic Intensity Zone | Peak Ground Velocity (PGV) $\left(\frac{cm}{s}\right)$ |
| --- | --- |
| Zone I | 60 < PGV < 80 |
| Zone II | 40 < PGV < 60 |
| Zone III | 20 < PGV < 40 |

**Table 13.** Value of EMPI vulnerability parameters ($V_P$) (Compiled based on [161]).

| Parameter | Value |
| --- | --- |
| Soft story | No; Yes |
| Heavy overhangs | No; Yes |
| Apparent quality | Good; Moderate; Poor |
| Short columns | No; Yes |
| Pounding effect | No; Yes |
| Topographic effect | No; Yes |

EMPI classifies the vulnerability of a building as good, moderate, and miserable [63]. The initial and vulnerability scores ($V_S$) based on story number 1 to 7 are presented in Table 14.

**Table 14.** Value range of EMPI concrete buildings' initial and vulnerability scores (Compiled based on [161]).

| Initial Scores | | Vulnerability Scores | | | |
| --- | --- | --- | --- | --- | --- |
| Zone I | [70, 90] | Short Column | −5 | Apparent Quality | [−15, −10] |
| Zone II | [80, 125] | Soft Story | [−15, 20] | Heavy Overhang | [−15, −10] |
| Zone III | [−15, 20] | Pounding Effect | [0, −3] | Topographic Effects | [0, −2] |

The above given score modifiers were assigned on the basis of the statistical studies of 454 buildings that were surveyed after the 1999 Düzce earthquake, in 2007 [80,159,160,166–168]. The selected performance score (PS), or the final score, of the buildings is determined by taking into account the $B_S$, $V_P$, and $V_S$ parameters, as seen in Equation (19).

$$PS = B_S - \sum V_P \cdot V_S \tag{19}$$

### 2.9.2. The RBTE-2019 Method

The RBTE-2019 method [164], which is a simplified technique for determining the regional seismic vulnerability distribution on the basis of a statistically significant number

of buildings, was proposed as an RVS methodology in Turkey. This technique includes forms for RC, masonry, and mixed-use buildings (e.g., buildings having structural systems made of various materials). Equation (20) is used to calculate the building performance score (PP) for individual buildings:

$$PP = TP + \sum_{i=1}^{N}(O_i \cdot OP_i) + YSP \tag{20}$$

where TP is the basic score; $O_i$ is the value of the negativity parameter; $OP_i$ is the score of the negativity parameter; and YSP is the score of the structural system. As a result of the RVS calculations, the regional performance score is calculated, and the priority regions are determined.

The detailed seismic risk assessment methodologies of the existing RC, masonry and mixed-use buildings are classified into three sections on the basis of TEC-2018 [169] with respect to the building height: (1) Low-rise; (2) Mid-rise; and (3) High-rise. For the detailed seismic risk evaluation of the existing buildings, a horizontal acceleration spectrum is used to implement the seismic risk assessment of low-rise and mid-rise RC buildings. However, a nonlinear dynamic time-history analysis is needed in order to demonstrate the seismic risk of high-rise RC buildings. A seismic risk assessment of the existing masonry and low-rise mixed buildings could be implemented by using the horizontal elastic acceleration spectrum.

## 3. RVS Methods Developed for Regional and Special Building Types

There are RVS methods that have been developed for regional and special building types in addition to the widely used RVS methods. The methods outlined for the regional and particular building types are presented below.

### 3.1. RVS Methodologies Developed Partially on the Basis of Previously Presented RVS Methods and Used Regionally

EMS-98 [11] and GNDT II have been combined and modified to generate a vulnerability index method for the Iranian RVS methodology by the State Organization of Schools Renovation, Development and Mobilization of Iran (SOSRI) for school buildings. The SOSRI provides simplified survey forms for seismic assessments via the vulnerability index (R) of the Iran method, as is shown in Table 15. The R is the product of two values, $R_i$ and $k_4$, as is shown in Equation (21) [102].

$$k_4 = 3.4a - 0.43 \tag{21}$$

where $k_4$ denotes the potential seismic demand measure that is based on the PGA; and a is the probable PGA value of the school's seismic region. On the basis of these calculations, demolition or retrofitting can be determined depending on the vulnerability index value of the building that is employed by the SORSI [142].

**Table 15.** Vulnerability index (R) of Iran (Compiled based on [102]).

| Seismic Vulnerability | | Decision |
|---|---|---|
| Low | $R \leq 25$ | No further detailed assessment required! |
| Moderate | $25 < R \leq 75$ | Further detailed assessment required! |
| High | $R \geq 75$ | Demolish and reconstruct! |

In order to calculate the seismic vulnerability of the selected urban area of Annaba City in Algeria, in a study by Athmani et al. [141], a damage evaluation and seismic vulnerability index methods (RISK-UE LM1 based on EMS-98 [11] and GNDT Level II) were adapted and applied to analyze the masonry buildings. The two selected methods have shown very good agreement in terms of representing the vulnerabilities of the considered buildings [141].

A seismic vulnerability assessment was carried out by Tyagunow et al. (2006) [170] for Germany, in cooperation with expert groups from GeoForschungsZentrum Potsdam and Universität Karlsruhe, under the umbrella of the Center for Disaster Management and Risk Reduction Technology, using EMS-98 [11]. As a result of the research carried out by Tyagunow et al. [170], the structural vulnerability has been explained by using four vulnerability classes (A to D) for residential buildings in Germany, compared to the six vulnerability classes (A to F) in EMS-98. As a result of the seismic assessment performed by Schwarz et al. [171] in Central Germany, using the EMS-98 methodology and seismic risk assessment technology based on geographic information systems (GIS) [172], it was concluded that the seismic risk assessment becomes efficient if it is combined with the necessary calibrations and instrumental examinations in the evaluation process.

On the basis of the JDPA (1990), Lee et al. [173] provide basic details on the seismic risk evaluation of existing RC Korean buildings in 2001. Then, in 2018, the seismic risk assessment of 14 RC buildings was assessed on the basis of the JBDPA method [110,112] to evaluate the seismic capacity of existing RC buildings in Korea, and a capacity comparison was made between the Korean and Japan RC buildings [174]. The study, which was conducted by Lee et al. [174], emphasizes the urgent need for the development of the Korean seismic retrofit scheme. In order to apply the JDPA methodology [110,111] to the Korean RC buildings, the structural index was modified and calculated on the basis of the ultimate horizontal strength and ductility [114]. Inoue et al. [175], in 2017, also considered the JBDPA method [112] for the seismic assessment of existing RC buildings, with some modifications for the Bangladesh conditions.

The LM1 RISK-EU approach used in Morocco is based on the damage matrices of EMS-98 by Cherif et al. [176–178] for building vulnerability assessments [140]. Moreover, the majority of the buildings in Morocco are RC structures [116].

Vallejo [179] used the RVS methodology of the modified ATC 21 [9] to determine the seismic risk assessment of high-rise and mid-rise buildings in Manila, the Philippines. Clemente et al. (2020) [180] used the FEMA P-154 [8] RVS methodology to determine the seismic risk assessment of 26 hospital buildings in Manila.

The GNDT method was used with some modifications that were based on the vulnerability parameters (14 instead of 11) for the vulnerability index calculation in Portugal by Ferreira et al. (2013) [181]. Another similar approach, by Kassem et al. (2020) [116], classifies the buildings into two vulnerability classes (A and B), in the old city center of Seixal in Portugal, on the basis of the EMS-98 scale, using the vulnerability index ($I_v$), and the vulnerability index formulation in this study, which are based on the GNDT Level II methodology.

The seismic vulnerability assessment in Barcelona, Spain, by Lantada et al. (2010) [182], used the RISK-UE framework with the GIS technique to display the obtained information on a seismic map [116].

The seismic risk assessment methodology of the Swiss standard by the Swiss Society of Engineers and Architects (SIA) [183] consists of three stages [159]. The first stage contains a rough estimation of the seismic risk assessment that is based on visual screening and technical drawings [184]. The second stage involves a detailed seismic risk assessment of some of the selected elements. The third stage includes strengthening the intervention [183].

"Building's Technical Condition Passport" is used for the structural certification for the Ukrainian buildings to gather the structural properties (e.g., geological site conditions, structural elements, covers and roofs, and insulating coatings) to perform the structural assessment [185]. Moreover, Dorofeev et al. (2014) [186] consider prefabricated slabs, eccentric arrangements of concrete columns, and confined masonry structures, and suggest designs and calculations. Dorofeev et al. (2014) [187] offer a methodology that consists of a three-level system for the seismic assessment of existing Ukrainian buildings.

Didier et al. (2017) [188] performed research, in which the seismic risk of the structures was assessed using the RVS form recommended by the Nepal Engineers' Association (NEA) following the 2015 Gorkha earthquake. The RVS form recommended by the

NEA is essentially a copy of the form developed by the National Society for Earthquake Technology-Nepal (NSET) and the Department of Urban Development and Building Construction (DUDBC) [189] in 2009, and it is based on the ATC-20 [190] and the ATC-20-2 [191]. It enables the immediate use of the data obtained via the NEA RVS smartphone application. Rupakheti and Apichayakul (2019) [192] suggested an RVS approach for structures in Nepal that utilized data from the 2015 Gorkha earthquake, and the ordinal regression approach in the Statistical Package for the Social Sciences (SPSS) [193], a statistical program. Various regression methods were employed in this study to select the best regression model. It is claimed that the RVS methodology is quite reliable for fitting the analytic outcomes.

Gentile and Galasso [124] established the "Indonesian School Programme to Increase Resilience (INSPIRE)" in 2018, with the rapid visual screening of RC school buildings. The INSPIRE index ($I_v$) was calibrated using the HAZUS MH4's five fragility curves and was applied to 85 RC school buildings. The $I_v$ is made up of two elements, as illustrated in Equation (22): the baseline score ($I_{BL}$), in Equation (23), and a performance modifier ($\Delta I_{PM}$), in Equation (24).

$$I_v = I_{BL} + \Delta I_{PM} \tag{22}$$

$$I_{BL} = \left( \frac{50 - 1}{P_{HAZUS, \, max} - P_{HAZUS, \, min}} \right) \cdot (P_{HAZUS} - P_{HAZUS, \, min}) + 1 \tag{23}$$

$P_{HAZUS, \, max}$ represents the maximum value of the probability of exceeding DS3; $P_{HAZUS, \, min}$ represents the minimum value of the probability of exceeding DS3; and $P_{HAZUS}$ represents the probability of exceeding DS3 on the basis of the PGA value.

$$\Delta I_{PM} = \frac{1}{2} \sum_{1}^{8} w_i \cdot SCORE_i \tag{24}$$

The weight and score of each parameter are symbolized as $w_i$ and $SCORE_i$, respectively. The earthquake safety of the structures in Indonesia was assessed by Wahyuni et al. (2017) [194] using FEMA P-154 [8], which is a widely used RVS approach. Pujianto et al. (2019) [195] conducted a rapid postearthquake visual assessment of school buildings in Lombok, Indonesia, using the Indonesian standard [196]. Haryanto et al. (2020) [76] utilized the FEMA P-154 [8] RVS technique to assess the seismic vulnerability of nine RC buildings in Indonesia.

### 3.2. RVS Methodologies Developed for Special Types of Buildings

The World Health Organization and the Pan American Health Organization have developed a safety index (SI) method for assessing hospital buildings that uses an RVS method and that takes the structural and nonstructural components into account [197,198]. Then, Perrone et al. (2015) [199] implemented the proposed methodology in two hospital buildings in different seismic zones in Italy, and improved the existing method's SI.

Many researchers have focused on the RVS system for the seismic evaluation of school buildings, as described below. For the structural and nonstructural elements, Lang et al. [200] suggest two separate indices [197]. The SAARC Disaster Management Center (SDMC) [201] also required a rapid evaluation of school buildings using the RVS procedure. The RVS approach was used to review 15 school buildings on the island of Lombok [195]. Ruggieri et al. [197] aimed to calculate a safety index (SI) by considering both the data in the literature and a large number of RC school buildings in Italy with respect to the methodology proposed by Perrone et al. (2015) [199]. "Integrated Rapid Visual Screening of Schools: A How-to Guide to Mitigate Multihazard Effects Against School Facilities" has been published by the National Institute of Building Sciences (NIBS) [202] to evaluate the vulnerability of school buildings.

Lucksiri et al. (2012) [203] suggest a new RVS methodology that applies the sidewalk survey concept, which is similar to the FEMA 154 scoring procedure [204], with consideration to single-family wood-frame dwellings that have plan irregularities. The plan irregularity Rapid Visual Screening (piRVS) methodology consists of two parts: (I) Building

the configuration parameters on the basis of the shape parameters and the opening-related parameters; and (II) Building a seismic response prediction [203]. The seismic performances of the building, which is based on the drift limits, are as follows: immediate occupancy, 1%, Grade 4; life safety, 2%, Grade 3; collapse prevention, 3%, Grade 2; with 10% for Grade 1 and an exceedance of 10% for Grade 0 [203].

## 4. Overview of Research Projects Concerning Comparison of RVS Methods

This section compares the RVS methods with other widely used RVS methods and/or postearthquake data to demonstrate their reliability and applicability. In addition, it also determines their adequacy by comparing them with the results obtained from the detailed seismic risk assessment methods.

### 4.1. Comparison of RVS Methods

In several investigations, the structural damage states were identified by employing more than one rapid visual screening (RVS) approach for the seismic risk assessments. In this context, the findings of the research using more than one RVS approach are presented below.

According to Harirchian et al. (2020) [63], the FEMA P-154 is not commercially feasible because it determines that the damage states are worse than they really are. However, the Turkish method, the Earthquake Master Plan of Istanbul (EMPI) and the Indian method (IITK-GGSDMA) produced more reliable results. It is recommended that the IITK-GGSDMA method should be tested to see whether it provides the same results in Indian buildings and whether it could be used on the basis of the results obtained.

According to Moseley et al. (2007) [108], the distinction between the Earthquake Planning and Protection Organization (OASP) technique, which was established on the basis of the initial FEMA methodology, and the FEMA technique, is that the building stock in Greece is different and that the OASP methodology was implemented for the buildings.

Tischer et al. (2011) [134] utilized the Canadian method, NRCC92, and FEMA 154 to evaluate roughly 100 school buildings in Montréal. The results corroborated that these two approaches were in a reasonable amount of agreement. The NRCC92 was developed on the basis of expert opinion; therefore, it is difficult to revise. However, FEMA 154 was developed on the basis of the capacity spectrum method; therefore, it is simple to modify and it can be applied to other countries. By predicting the worst possible scenario in the FEMA 154 methodology, excessively conservative results are obtained. On the other hand, seven different irregularities were defined in the NRCC92 to properly consider the shortcomings.

Athmani et al. (2015) [141] stated that the results obtained with the Italian method (GNDT II) and the European method (RISK-UE LM1) were identical, and the reason for the slight variation was the difference in the statistical functions used in each approach.

There were variations in the weights assigned to the various parameters in the assessments using the five methodologies by Bhalkikar and Pradeep (2021) [163] for Agartala City, which resulted in diverse outcomes. Since each RVS technique has a distinct scoring system, it is argued that comparing RVS systems is difficult. In this context, the RVS methodologies were compared to the damage grades that were acquired from the seismic assessment. Furthermore, it should be mentioned that a numerical study of buildings should be performed in order to enhance the RVS approaches.

According to Calvi et al. (2006) [5], the RVS techniques consider the various uncertainties in different ways to date, and it is recommended that a combination of the positive characteristics of the various (at least two) techniques should be used for a reliable vulnerability assessment in a particular location in the future. Furthermore, the EMS-98 damage grades were revised on the basis of the postearthquake data because of Achs and Adam's (2011) [205] assessment with regard to historical brick masonry residential buildings in Vienna, Austria. Achs and Adam (2011) [205] employed postearthquake data in order to improve the traditional RVS methodologies.

Despite the challenges of comparing various RSV techniques because of their diverse scoring systems and design strategies, they were carried out as described above. Even though FEMA 310 indicates that FEMA 154 should be implemented first, FEMA 154 was later updated twice. Therefore, comparing FEMA 154 to FEMA 310 in light of the current improvements can demonstrate the effectiveness of the modifications. Although the NZSEE is based on the FEMA 154 RVS method, it differs in that it takes into account the building plan area, the span-to-depth ratio, as well as other factors when calculating the attribute score. Accordingly, it is simple to perform the necessary calculations to modify it, depending on the obtained findings, because the FEMA method was developed using the capacity spectrum technique. On the other hand, the NRCC method is difficult to modify because it was developed on the basis of expert opinion. Since the GNDT approach was designed for postearthquake building evaluation, it needs to be altered for pre-earthquake screening before comparing it with other methods in terms of the damage identification capabilities. The JDPA, which is employed for RC buildings, is the approach that is considered to provide the most comprehensive building information from the methods described above.

On the other hand, the RVS method must be implemented to determine whether a building is habitable after an earthquake, when the safety level of the building depends on a rapid assessment. On the basis of the pre-earthquake screening and the postearthquake data of the buildings damaged in the Tirana earthquake, our research [206] suggests that the EMS method is simple and also adequate for justifying the habitability of a building after an earthquake. However, building inspection forms for the EMS method need to be designed in order to be able to examine the buildings before an earthquake. As mentioned in the "EMS-98 Scale" section, attempts have been made to resolve the contained uncertainties in the subsequent research [12,85] by employing fuzzy logic.

The authors [206] also compared the postearthquake building screening data collected after an earthquake that hit Albania in 2019 with the FEMA P-154 [8] RVS method. The results with the application of the FEMA P-154 [8] RVS method and the postearthquake data did not display adequate agreement. Moreover, the authors compared FEMA P-154 [8] to FEMA 154 [10]. This comparison shows that the scores obtained by using the two methods did not always correlate, and that FEMA P-154 [8] was significantly more conservative than FEMA 154 [10]. Therefore, fuzzy logic and machine learning algorithms were used by the authors to improve the current methods with regard to the postearthquake building screening data, or to design a new method with a self-development capability on the basis of the available data.

### 4.2. Comparison of RVS with Detailed Vulnerability Assessment

In certain conditions, an RVS technique needs to be designed or adapted on the basis of the structural characteristics and the site seismicity of a particular region. New RVS methods could be developed and/or enhanced by considering the findings of the DVA (detailed vulnerability assessment) implementation, or the existing RVS techniques may be modified. Next, the results on developing a new RVS approach and/or adapting an existing RVS method are summarized.

Usually, the DVA is used for the RVS method design, as has been described above. According to Kumar et al. (2017) [159], a preliminary DVA of the selected buildings should be performed in order to calibrate the RVS technique.

Ruggieri et al. (2020) [197] demonstrate the adequacy of their recent RVS approach for RC school buildings on the basis of a new safety index by taking into account the seismic risk parameters (hazard, vulnerability, exposure), with a nonlinear static analysis.

Teddy et al. (2016) [207] utilized the static pushover analysis, which is a detailed analysis methodology, to verify the reliability of the RVS procedure that was recommended by the FEMA in 2015. This study concluded that the RVS approach is quite suitable for the seismic performance prediction (used as a result of comparing FEMA 2015 to the static pushover analysis approach in the cases of six structural models).

Although the National Research Council of Canada (NRCC) methodology was created for the Canadian building stock, it lacks a detailed assessment [80]. To date, it has been recommended that this gap be addressed with a hybrid technique (based on a combination of FEMA 310 [68] and the Indian standard, IITK-GSDMA [150]), which is recommended by Alam et al. (2012) [80].

The vulnerability index method (VIM) was derived for Italy by using large amounts of damage data [208]. Barbat et al. (2010) [208] state that the VIM and the capacity spectrum methods correlate well with the key elements of the inhabited environment of Barcelona.

Wang (2007) [209] performed an RVS assessment that was based on the local maximum considered earthquake (MCE). The MCE considers that there will be an earthquake, with a 2% probability of exceedance, in 50 years, and it is developed by considering one or more earthquake data for a wide variety of buildings. The site-specific MCE values, less than the median value, cause the overestimation of the MCE values and the corresponding seismic hazard, whereas the values that are more than the median value cause an underestimation. Therefore, a final score obtained for a building on the basis of RVS might be incorrect. Adjustments should be made in order to avoid the systematic errors that may arise if the local MCE value is used while implementing an RVS methodology.

Furthermore, in certain circumstances (because of a lack of time, money, tools, or data), it is necessary to examine a large building stock and to rapidly identify which buildings needed to be retrofitted on the basis of the screening instead of the DVA methods. Thus, the RVS methods can be utilized to identify the necessity of retrofitting buildings [2,67,102,118,210,211].

## 5. Discussion on RVS Methodologies

The seismic standards are developed on the bases of the experiences and lessons learned from the earthquakes that have occurred in earthquake-prone areas. Moreover, the lessons learned from previous earthquakes could be used to improve the current RVS methods, or to modify other methods so that they can be employed in different places. For example, data from previous earthquakes were utilized in the development of the GNDT [138] RVS method in Italy, and FEMA 154 (1988) [9] was modified to develop FEMA 154 (2002) [10] because of the lessons learned from earthquakes in the 1990s. In this context, the parameters (e.g., vertical and plan irregularities as shown in Table 16) that decrease the earthquake resistance of structural systems need to be eliminated, since this has been learned from previous earthquakes and is considered in the current RVS methods.

In regions that are in need of assessments for the built environment against earthquakes, the lack of such RVS methods is a problem; however, the current RVS methods should also be adjusted to be implemented for other locations. The major causes for the changes are the distinct features of the building stocks in the various locations, such as the material utilized, the type of structure constructed, and the soil conditions and the seismicity of the area.

Unfortunately, because of the long return period of a strong earthquake in low- or moderate-seismic regions, the current standards cannot be updated with the lessons learned and the experience gained from earthquakes in the area in which they are employed. This leads to present regulations and implementations that ignore the seismic loads and the parameters that decrease the building seismic resistance until a major earthquake happens [2]. Thus, improvements should be performed in areas with low or medium seismicity by incorporating the valuable lessons learned from earthquakes in high-seismic regions, by modifying the current RVS methods and/or adopting from other methods.

Recently, some researchers [206,212–214] have implemented the existing traditional RVS methodologies in specific regions. Generally, they state that further research is needed to implement or modify the method, and that this methodology needs to be developed on the basis of the pre- or postearthquake data, the detailed seismic risk assessment (DSRA) methodologies, and/or the Soft RVS (S-RVS) methodologies.

RVS methods are mostly validated by utilizing various techniques. Section 3.2 presents the investigations carried out for the enhancement of the traditional RVS techniques using DSRA methodologies. The DSRA techniques are divided into elastic and nonlinear analysis methods, which consist of the nonlinear static analysis (e.g., capacity spectrum analysis method, pushover analysis, N2 method), and the nonlinear dynamic analysis (e.g., incremental dynamic analysis (IDA), endurance time method). The IDA method, which was developed on the basis of nonlinear dynamic analyses for a more realistic performance calculation of a building, does not require the first mode of a building to be dominant, and it considers the contribution of each mode of the building. Moreover, the collapse mechanism of a building could be examined more accurately by creating a precise structural analysis model by applying the finite element method (FEM) and the applied element method (AEM), which are extensively utilized in the literature.

Other possibilities for enhancing the RVS methodologies are the S-RVS methods, which are based on fuzzy logic, artificial neural networks, machine learning, and deep learning. S-RVS methods can be developed by employing computational algorithms, which are trained by utilizing postearthquake data and/or expert-opinion-based data. By comparing the S-RVS findings with the separated data for the reliability (test) checks, the accuracies of the developed S-RVS techniques are determined.

When the RVS method is compared to the DSRA technique in terms of the time required to perform the structural assessment, the RVS assessment technique may be completed faster compared to any time-consuming DSRA techniques.

This paper provides a broad review of the traditional RVS methodologies. Numerous techniques have been developed around the globe and a comparison of these techniques is presented on the basis of the highlighted results from the literature. Hence, establishing a relationship between these techniques have been challenging because of the fact that the methods use different parameters and have been developed for different materials, structural properties, or for different building types or areas [215]. Table 16 illustrates the considered parameters by a variety of RVS techniques. However, we have presented the main differences between the RVS methods to provide an overview of the details and the possible accuracy of these procedures.

The current RVS methods differ from one another in terms of the pre- (e.g., FEMA 154, NRCC, NZSEE, etc.) or postearthquake (e.g., GNDT, EMS) screening capabilities and development techniques (e.g., expert opinion, fuzzy logic, capacity spectrum, and fragility curves). Some methods need to be modified (e.g., GNDT, EMS) before being used for pre-earthquake building screening since they were intended to be used for postearthquake building screening. Postearthquake building damage states can easily be diagnosed on the basis of the observable damage; however, determining this before an impending earthquake is challenging. Therefore, the most suitable RVS method for evaluation is decided on the basis of the characteristics of the different RVS methods. Although the RVS methods are not computationally expensive (e.g., FEMA 154 takes 15 to 30 min per building), they do require the participation of a large number of trained personnel, and they take a long time when they are used to examine a building stock. Moreover, despite the fact that the JDPA may provide more accurate results for RC buildings, it takes significantly more time than other methods, such as FEMA 154 and NRCC. Furthermore, to make the methods more accurate, it is necessary to eliminate the vagueness and uncertainty that arise during their application. Therefore, the data from the postearthquake screening and the lessons learned from the earthquakes are critical for the development of these methods. Improving the accuracy of the methods would result in more effective decisions for insurance companies, urban planning, and seismic mitigation programs.

**Table 16.** Comparison of parameters in different RVS methods.

| | | | USA | | European Union | | Japan | New Zealand | Greece | Canada | Italy | India | Turkey | |
|---|---|---|---|---|---|---|---|---|---|---|---|---|---|---|
| | | | FEMA P-154 [8] | FEMA 310 [68] | EMS-98 Scale [11] | RISK-UE Project [12] | JBDPA [109–113] | NZSEE [123] | OASP [13] | NRCC [14] | GNDT [138] | IITK-GSDMA [150] | EMPI [161] | RBTE-2019 [164] |
| Vulnerability Parameters | Building Irregularities | Vertical Irregularity | ✓ | ✓ | ✓ | ✓ | ✓ | ✓ | ✓ | ✓ | ✓ | ✓ | ✓ | ✓ |
| | | Plan Irregularity | ✓ | ✓ | ✓ | ✓ | ✓ | ✓ | ✓ | ✓ | ✓ | ✓ | ✓ | ✓ |
| | | Pounding Effect | ✓ | ✓ | ✓ | × | × | ✓ | ✓ | ✓ | × | ✓ | ✓ | ✓ |
| | Basic Building Information | Structure Type | ✓ | ✓ | ✓ | ✓ | ✓ | ✓ | ✓ | ✓ | ✓ | ✓ | ✓ | ✓ |
| | | Foundation Type | × | ✓ | × | ✓ | ✓ | ✓ | × | ✓ | × | ✓ | × | x |
| | | No. of Story | ✓ | ✓ | ✓ | ✓ | ✓ | ✓ | ✓ | ✓ | ✓ | ✓ | ✓ | ✓ |
| | | Construction Year | ✓ | ✓ | × | ✓ | ✓ | ✓ | ✓ | ✓ | ✓ | ✓ | ✓ | ✓ |
| | | Occupancy | ✓ | ✓ | ✓ | ✓ | ✓ | × | ✓ | ✓ | ✓ | ✓ | × | ✓ |
| | Building Technical Information | Prior Strengthening | × | × | ✓ | ✓ | × | ✓ | ✓ | × | ✓ | × | × | x |
| | | Prior Damage | × | ✓ | ✓ | × | ✓ | × | ✓ | × | ✓ | × | × | x |
| | | Maintenance | × | ✓ | ✓ | ✓ | × | × | ✓ | × | ✓ | ✓ | ✓ | ✓ |
| | | Pre-Code | ✓ | × | ✓ | ✓ | × | ✓ | ✓ | ✓ | × | ✓ | × | x |
| | | Post-Benchmark | ✓ | ✓ | ✓ | ✓ | × | × | × | ✓ | × | × | × | x |
| | | Falling Hazards | ✓ | × | × | × | × | × | × | ✓ | ✓ | ✓ | ✓ | x |
| | Site Seismicity and Soil Characteristics | Site Seismicity | ✓ | ✓ | ✓ | ✓ | × | ✓ | ✓ | ✓ | × | ✓ | × | x |
| | | Soil Type | ✓ | ✓ | ✓ | ✓ | × | ✓ | ✓ | ✓ | ✓ | ✓ | × | ✓ |
| | | Liquefaction | ✓ | ✓ | × | × | × | ✓ | × | ✓ | × | ✓ | × | x |

Note: ✓: considered; ×: not considered.

Although traditional RVS methods are capable of screening and identifying the seismic reliability of existing buildings in a relatively short time, they have some flaws (based on the site-specific characteristics, the surveyor bias, and the data uncertainty and vagueness in the evaluation). Thus, various methodologies can be used to improve the existing RVS methods or to develop new ones. Simplified structural analysis models (e.g., single-degree-of-freedom or multi-degree-of-freedom lumped mass structural systems) can be created to analyze many buildings and to calibrate and/or develop the RVS methods in future research and development. The accuracies of the applied methods could be determined by conducting virtual pre-earthquake screenings of the buildings using a software, such as Google Earth, and comparing the results with the postearthquake data [206]. As an alternative, the data collected through the postearthquake building screening could be utilized in computer algorithms, such as machine learning, fuzzy logic, and neural networks, in order to enhance the existing RVS methods and/or to develop new ones. In addition, GIS may be used to collect building data, such as the building pictures and location. Following that, the collected digital pictures can be examined using image processing algorithms to identify the RVS parameters, such as the plan and vertical irregularity, the building height, the construction quality, and the pounding. The obtained building damage state information can also be shown on a GIS-based map.

We have aimed to offer a synopsis of the existing RVS approaches. It was also our intention to assist with the determination of an appropriate method to serve as the basis for further development, which applies fuzzy logic and other mathematical solutions to overcome the bias of surveyors, the uncertainty in the data, and the vagueness in the evaluation. In addition, a review of the current S-RVS methodologies used in the development of the RVS methods has been briefly explained [206].

## 6. Conclusions

The structural safety of existing buildings needs to be assessed since some of them might be at risk, and impending earthquakes may cause economic losses. Therefore, this study presents a state-of-the-art review of the traditional rapid visual screening (RVS) methodologies for the seismic vulnerability assessments of existing buildings. Each method and their development have been explained briefly. Evaluations have been presented and discussed on the basis of the findings of previous studies (see Sections 4.1 and 4.2). The comparison has a predecision feature in the selection and application of the RVS methods. The objective of this study was to provide an overview of the conventional RVS approaches that might be utilized in future investigations by a broad audience, such as engineers, architects, and insurance companies. Although the comparison of the various RVS techniques has been challenging, this study describes, evaluates, and compares the results of the studies on the traditional RVS methods, and it also reveals the preferred methods to select for a specific purpose. Moreover, the accuracy levels of the methods and their application areas (e.g., for the pre-earthquake NZSEE, FEMA 154, and IITK-GSDMA, and the postearthquake GNDT and EMS) have been explained, along with the evaluation of the different approaches in terms of the necessary time for the evaluation of a building. The vulnerability evaluation of buildings can be performed conveniently using the approaches described above. Finally, one of the main advantages of the presented techniques is that it may be employed to assess buildings in different regions, with appropriate adaptations. In addition, regional seismic risk scenarios could also be developed by employing RVS techniques.

However, different RVS approaches might produce distinct results when assessing seismic risks. These discrepancies are due to their considering unique parameters, with the impact of each parameter varying in each methodology. Consequently, the current RVS techniques could be calibrated, or new RVS methods could be developed. Therefore, the existing conventional RVS procedures should be well known so that the appropriate modifications and/or developments can be performed to overcome the deficiencies (on the basis of the site-specific characteristics, surveyor bias, data uncertainty, and vagueness

in the evaluation). (1) To improve the conventional RVS methods, a variety of characteristic buildings could be analyzed using simplified seismic assessment methods; (2) The findings of the pre-earthquake virtual screening, using a software such as Google Earth, can be compared to the postearthquake screening data to illustrate the accuracies of the RVS methods; (3) Fuzzy logic and machine learning algorithms can be used with the postearthquake building screening data to overcome the shortcomings of the traditional RVS methods. By applying these techniques, adjustments could be made to each parameter utilized, in addition to a significant modification in the RVS method. The influence of each considered parameter on the performances of the buildings should be thoroughly understood, and their relative weights should be fixed in order to make the RVS evaluation more reliable.

**Author Contributions:** Conceptualization, N.B. and O.K.-B.; investigation, N.B.; writing—original draft preparation, N.B.; writing—review and editing, O.K.-B.; supervision, O.K.-B. All authors have read and agreed to the published version of the manuscript.

**Funding:** This research received no external funding.

**Institutional Review Board Statement:** Not applicable.

**Informed Consent Statement:** Not applicable.

**Data Availability Statement:** Not applicable.

**Conflicts of Interest:** The authors declare no conflict of interest.

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
