# Peer review of "Conventional RVS Methods for Seismic Risk Assessment for Estimating the Current Situation of Existing Buildings: A State-of-the-Art Review"

_sustainability, doi:10.3390/su14052583_

Round 1

Reviewer 1 Report

The authors reviewed the conventional RVS methods of seismic risk assessment for estimating current situation of existing buildings. This is an interesting and useful topic. The authors did a very detailed summary of the RVS methods in various codes in different countries and regions. Several comments should be addressed as follows:

  1. Some sentences are confusing, e.g., line 910-911, RVS methods are also compared with other generally accepted RVS methods…
  2. There are a lot of citation errors.
  3. Line 92-102: All of them are Section 0.
  4. The content of Section 3 are too limited.
  5. The second Section 4.1 should be modified to 4.2.
  6. There should be more discussion and comparisons between different RVS methods.

Author Response

We would like to express our gratitude to Reviewer – 1 for his/her time and significant feedback. The modifications and/or additions made in response to the given comments are explained one by one in the attached file.

Reviewer 2 Report

This study presents a state-of-the-art review of traditional rapid visual screening (RVS) methodologies for seismic vulnerability assessment of existing buildings. The paper should improved considering following comments before acceptance.

  1. A general review in the language of the manuscript could provide a better understanding for readers. I think some parts of the manuscript are literal translation from the primary language. Hence, they are difficult to understand. At least, the reader should try hard to understand these sentences. Hence, a comprehensive language revision is needed. I suggest revising the manuscript by a native english speaker.
  2. Some sections are very detailed. Shortening is recommended.
  3. Introduction of the manuscript should be more well prepared. More general literature review is needed and novelty of the research should be stated clearly. The importance of the presented work is not clarified. In other words, the novelty and importance of the research should be clarified using up-to-date researches around the globe. Experiences from all over the world could help authors for performing a comprehensive research.
  4. Introduction: what is the question of the present study? what is the contribution for the knowledge? Please give more information about RVS methods. Besides, the following papers can be utilized in introduction section.
  • Aksoylu, C., Mobark, A., Hakan Arslan, M., & Hakkı Erkan, İ. (2020). A comparative study on ASCE 7-16, TBEC-2018 and TEC-2007 for reinforced concrete buildings. Revista de la construcción, 19(2), 282-305.
  • Aksoylu, C., & Kara, N. (2020). Strengthening of RC frames by using high strength diagonal precast panels. Journal of Building Engineering, 31, 101338.
  1. The work is a good report of RVS investigation, but which are the lessons learnt? The authors have to clarify before acceptance.

Author Response

We would like to express our gratitude to Reviewer – 2 for his/her time and significant feedback. The modifications and/or additions made in response to the given comments are explained one by one in the attached file.

Reviewer 3 Report

This paper offers a comprehensive review of rapid visual screening (RVS) methods for seismic assessment. The scope of the paper is of importance to the earthquake engineering community and presents a comparative assessment of RVS methods employed in different countries and codes. Writing review papers are quite challenging, and as such, I appreciate the authors’ effort in creating a guide for other researchers. However, while the paper compiles some helpful literature, it does not organize data clearly and concisely. Therefore, the paper reads similar to listing several papers and guidelines than helping the readers understand the relationship between different kinds of literature, how they are connected or different, etc. Therefore, I suggest that authors examine new ways to organize literature to communicate “insights” about them (e.g., similar to the authors in Table 19). For example, authors can create a timeline graphic to quickly communicate the progress in RVS codes for chronological literature.

Additional comments to further improve the manuscript are provided below:

Technical comments

  1. The paper often lacked a critical perspective regarding cited literature and codes. The authors sometimes demonstrated how RVS methods are different (e.g., lines 582-590 Page 17) but did not provide a critique of those methods. It would be very insightful to discuss the limitations of described RVS methods.
  2. It is strongly suggested to discuss future research needs (based on the surveyed literature) to employ RVS methods. This discussion should be backed by the authors’ critique when reviewing different literature.
  3. The authors briefly mentioned that RVS is a method for fragility assessment. However, since a broad literature exists on other methods to assess building fragilities based on performance-based engineering, it is recommended that the authors briefly discuss other methods to derive fragilities. In addition, please also cite some relevant literature regarding determining seismic fragility using other methods than RVS as follows:

[1] Burton, H. V., & Deierlein, G. G. (2018). Integrating visual damage simulation, virtual inspection, and collapse capacity to evaluate post‐earthquake structural safety of buildings. Earthquake Engineering & Structural Dynamics47(2), 294-310.

[2] Zaker Esteghamati, M., & Farzampour, A. (2020). Probabilistic seismic performance and loss evaluation of a multi-story steel building equipped with butterfly-shaped fuses. Journal of Constructional Steel Research172, 106187.

[3] Baker, J. W. (2015). Efficient analytical fragility function fitting using dynamic structural analysis. Earthquake Spectra31(1), 579-599.

[4] Zaker Esteghamati, M., & Flint, M. M. (2021). Developing data-driven surrogate models for holistic performance-based assessment of mid-rise RC frame buildings at early design. Engineering Structures, 245, 112971.

[5] Silva, V., Akkar, S., Baker, J., Bazzurro, P., Castro, J. M., Crowley, H., ... & Vamvatsikos, D. (2019). Current challenges and future trends in analytical fragility and vulnerability modeling. Earthquake Spectra35(4), 1927-1952.

  1. I suggest that the authors add additional motivation and explanation about the purpose and outcome of this literature review in both the introduction and conclusion. Particularly, authors should clearly state the target audience and what the authors expect the audience to gain by reading this paper.
  2. On Page 3, the authors discussed how different nonlinear analyses are performed at the third assessment stage. As each of the discussed methods is rooted in rich literature, I strongly suggest adding a brief explanation and citing relevant literature. At a minimum, the authors should briefly discuss IDA, ET, and pushover analysis methods. Please check these references for examining how each method is used in the seismic analysis:

[1] Pang, Y., & Wang, X. (2021). Enhanced endurance-time-method (EETM) for efficient seismic fragility, risk and resilience assessment of structures. Soil Dynamics and Earthquake Engineering, 106731.

[2] Zaker Esteghamati, M., Banazadeh, M., & Huang, Q. (2018). The effect of design drift limit on the seismic performance of RC dual high‐rise buildings. The Structural Design of Tall and Special Buildings, 27(8), e1464.

[3] Gentile, R., & Galasso, C. (2021). Simplicity versus accuracy trade-off in estimating seismic fragility of existing reinforced concrete buildings. Soil Dynamics and Earthquake Engineering144, 106678.

[4] Zaker Esteghamati, M., Lee, J., Musetich, M., & Flint, M. M. (2020). INSSEPT: An open-source relational database of seismic performance estimation to aid with early design of buildings. Earthquake Spectra36(4), 2177-2197.

[5] Miano, A., Jalayer, F., Ebrahimian, H., & Prota, A. (2018). Cloud to IDA: Efficient fragility assessment with limited scaling. Earthquake Engineering & Structural Dynamics47(5), 1124-1147.

Editorial comments

  1. I do not understand why authors used a table format to communicate a single column of data throughout the paper. I suggest that the authors revise this format for communicating information. For example, Table 1 can be easily communicated as a list in the manuscript text, Tables 3 & 4 can be combined in a single table that presents both data, and so on.
  2. Some minor English editing is needed.
  3. None of the tables and figures are cited in the manuscript text. This is a fundamental flaw as it is unclear how tables and figures relate to the manuscript text. I believe the reason behind this flaw is that the authors used an automated system to create references to tables and figures, which generated errors in several places such as line 51 Page2, line 62, Page 2, etc.
  4. Some writing style editing is needed. For example, In line 38 Page 1, a comma is missed, and the abbreviation after Rapid Visual Screening needs to be added. Page 2 line 45, too many vague words are used without clear references (“them” and “they” refer to the buildings in the previous line). There are other instances that I did not list here for brevity.
  5. Page 3 lines 90-102, all the sections are named as 0. please fix that. There are other places in the manuscript where the section number is reported as 0 as well.

Author Response

We would like to express our gratitude to Reviewer – 3 for his/her time and significant feedback. The modifications and/or additions made in response to the given comments are explained one by one in the attached file.

Round 2

Reviewer 3 Report

The authors have adequately addressed all my comments and as such.